# A Glycolipidated-liposomal peptide vaccine confers long-term mucosal protection against *Streptococcus pyogenes* via IL-17, macrophages and neutrophils

Victoria Ozberk[1,3], Mehfuz Zaman[1,3], Ailin Lepletier [1], Sharareh Eskandari [1], Jacqualine Kaden[1], Jamie-Lee Mills [1], Ainslie Calcutt [1], Jessica Dooley[1], Yongbao Huo[1], Emma L. Langshaw[1], Glen C. Ulett [2], Michael R. Batzloff[1], Michael F. Good [1] ✉ & Manisha Pandey [1] ✉

Mucosally active subunit vaccines are an unmet clinical need due to lack of licensed immunostimulants suitable for vaccine antigens. Here, we show that intranasal administration of liposomes incorporating: the *Streptococcus pyogenes* peptide antigen, J8; diphtheria toxoid as a source of T cell help; and the immunostimulatory glycolipid, 3D(6-acyl) PHAD (PHAD), is able to induce long-lived humoral and cellular immunity. Mice genetically deficient in either mucosal antibodies or total antibodies are protected against *S. pyogenes* respiratory tract infection. Utilizing IL-17-deficient mice or depleting cellular subsets using antibodies, shows that the cellular responses encompassing, CD4⁺ T cells, IL-17, macrophages and neutrophils have important functions in vaccine-mediated mucosal immunity. Overall, these data demonstrate the utility of a mucosal vaccine platform to deliver multi-pronged protective responses against a highly virulent pathogen.

Group A streptococcus (*Streptococcus pyogenes*) infections and their sequelae are important causes of morbidity and mortality. Global mortality estimates for *S. pyogenes*-associated diseases including acute rheumatic fever (ARF) and rheumatic heart disease (RHD) exceed 500,000 deaths per year[1]. *S. pyogenes* colonizes the epithelial surfaces of the upper respiratory tract (URT) and skin, from where it can progress to invasive and immune-mediated diseases. There is an estimated incidence of 616 million cases of *S. pyogenes* pharyngitis and 111 million cases of *S. pyogenes* pyoderma per year[1]. Invasive streptococcal disease, including streptococcal toxic shock syndrome, has a mortality rate of 7–23%[2].

A vaccine is urgently needed. However, there are two major hurdles. Firstly, the pathophysiology of ARF and RHD is thought to be driven by autoimmune molecular mimicry between *S. pyogenes*

antigens cross-reacting with human-tissue[3]. M-protein cross-reactive epitopes have been identified in the B-repeat region of the protein[4]. Secondly, antigenic strain variation is extensive[5], thus challenging even a multivalent vaccine approach. A minimal conserved epitope approach to vaccine development addresses both of these concerns.

We defined a cryptic B cell epitope, J8, from the highly conserved C3-repeat region of the M-protein[6]. J8 can be conjugated to a carrier protein, such as diphtheria toxoid (DT) or Cross-Reactive Material-197 (CRM), to render it immunogenic in an outbred population[7]. Vaccination with J8-DT adjuvanted with Alum resulted in protection against skin disease[8] and J8-DT/Alum was demonstrated to be safe and immunogenic in both a rat model of autoimmune valvulitis[9] and in a randomized pilot clinical trial[10]. However, this vaccine was not protective against streptococcal mucosal infection[11]. Parenteral

[1]Institute for Glycomics, Griffith University, Gold Coast, QLD, Australia. [2]School of Pharmacy and Medical Science, and Menzies Health Institute Queensland, Griffith University, Gold Coast, Australia. [3]These authors contributed equally: Victoria Ozberk, Mehfuz Zaman. ✉e-mail: michael.good@griffith.edu.au; m.pandey@griffith.edu.au

immunization with immunodominant but polymorphic amino-terminal epitopes of the M-protein can induce immunity in the URT[12], but a vaccine capable of inducing strain-transcending immunity in the URT is required. Vaccination with liposomes containing J8[11,13] or with proteasomes containing the closely related peptide, J14[14], led to a short-term mucosal IgA response and protection against challenge infection. However, the correlates of immunity were not defined and strategies to improve the level and durability of immunity are needed to empirically design vaccine candidates.

It is widely assumed that IgA is responsible for mucosal protection[11,14,15]. Indeed, IgA specific for the M-protein, but not IgG, can provide passive mucosal immunity to mice[15] and M-protein-specific human IgA decreases adherence of streptococci to pharyngeal cells[16]. However, these studies do not exclude an important role for other mechanisms of streptococcal immunity. It was shown that strepto-coccal mucosal infection leads to interleukin (IL)−6 and IL-17 production by splenocytes and that CD4[+] T cells from IL-17-competent mice, but not from IL-17 deficient (*Il17*[−/−]) mice, can passively transfer resistance in some mice to mucosal infection[17].

Innate immunity is also important in defence against *S. pyogenes*[18]. Depletion of macrophages during the early phase of *S. pyogenes* infection led to decreased survival in mice in comparison to non-depleted controls[19]. The absence of macrophages was also associated with the dissemination of bacteria into the blood and internal organs[20]. Mice deficient in tumor necrosis factor (TNF) failed to recruit macrophages, but not polymorphonuclear leukocytes (PMNs), to the site of infection and were susceptible to *S. pyogenes* challenge[20]. Macrophages can be activated by T cells and provide non-opsonic phagocytosis[21,22] and they can also work with antibody to provide opsonic phagocytosis against infectious organisms[23]. Macrophages have been shown to play an important role in antibody-dependent J8-DT-mediated skin protection[8]. They thus have key roles in both innate and adaptive immunity.

To improve vaccine-induced mucosal immunity, we asked whe-ther a toll-like receptor 4 (TLR4) agonist would augment the level and durability of vaccine-mediated protection. TLR4 agonists are widely used in modern vaccines to increase T-helper (Th)−1 responses and to broaden the class of antibodies that are induced[24,25]. They have also been reported to augment Th17 responses[26]. 3D(6-acyl) PHAD (PHAD; also known as GLA) is a synthetic analog of detoxified monophos-phoryl lipid A (MPLA). MPLA is a TLR4 ligand that comprises the lipid A portion of *Salmonella minnesota* R595 lipopolysaccharide (LPS; endotoxin)[27]. TLR4 stimulation leads to myeloid differentiation factor 88 (MyD88) signaling and activation of NF-κB and downstream expression of pro-inflammatory cytokines, such as TNF[28], IL-6[29], IFN-γ[30] and IL-17[31]. Expression of these cytokines enhances the adaptive immune response by stimulating the maturation of antigen-presenting cells (APCs)[32]. MPLA has been used in human vaccine trials where it was shown to enhance antibody responses and polyfunctional T cell responses[33]. It was successfully utilized with a suspension of liposomes in a GlaxoSmithKline human-applicable delivery system, AS01[34]. It showed promising safety and efficacy in children and infants and is part of licensed vaccine formulations[35,36]. The ability of TLR4-activating adjuvants to induce or augment a mucosal IgA response has been demonstrated in two studies. In the first study, a liposomal vaccine construct was designed to express the *S. pyogenes* vaccine antigen J8 and influenza A virus antigen M2e on its surface[13]. The vaccine was adjuvanted with the inclusion of PHAD[13]. This combinatorial vaccine was able to induce an IgA response and protect mice from both pathogens[13]. The second study demonstrated that intranasal delivery of a TLR4-activating adjuvant (GLA-SE) switches the CD4[+] T cell response in the lungs from a Th1 to a Th17-dominated tissue resident response[37].

Here we describe a self-adjuvanting mucosal vaccine delivery system consisting of liposomes encapsulating DT as a source of T cell

help and displaying lipidated J8 (to facilitate insertion into the lipo-some membrane) and assess the ability of PHAD to enhance immunity. We test the persistence of vaccine-mediated antibody and cellular memory responses and use gene knock-out mice and depletion studies to assess the roles of antibody and adaptive cellular responses in PHAD vaccine-mediated mucosal immunity.

## Results

### An overview of *S. pyogenes* (2031) URT infection in mice
We compiled data from 6 different URT challenge experiments with *S. pyogenes* 2031 (an *emm* 1 isolate from the Menzies School of Health Research, Darwin, Australia) to analyze the natural progression of *S. pyogenes* infection in BALB/c mice following URT challenge. Nasal shedding (NS) and throat swabs (TS) were collected on days 1–3 post-challenge and data were combined from all three days and represented as the geomean cfu ± geometric SD (Supp. Fig. 1). Bacterial enumera-tion from NS and TS are qualitative in nature and were found to be highly variable. The most accurate measurement of bacterial burden is quantification of the bacterial load from whole tissues because the entire tissue is excised, total bacterial load determined, and therefore was followed for all studies described in this paper (Supp. Fig. 1). The most important tissue following URT challenge is the Nasal Associated Lymphoid Tissue (NALT; mouse equivalent of human tonsils). The more severe the infection the more likely *S. pyogenes* will dissipate from NALT to lungs, spleen, and blood (sepsis)[38]. Bacterial counts from the NS and TS were 10–10,000-fold lower than from the NALT and lungs and likely came into the nasopharyngeal cavity following escape from the infected NALT or lungs, rather than from independent replication in the cavity[38]. Given that there is a high bacterial burden in the lungs of intranasally (i.n.) challenged mice, this model can equally be considered to be a lower respiratory tract model as well as a URT model.

We chose to examine the effect of vaccination on the NALT and lungs. We examined the contribution of the TLR4 agonist, the glyco-lipid PHAD, on vaccine efficacy and the duration and nature of vaccine-mediated protection.

### J8-Lipo-DT-PHAD immunization offers robust mucosal immunity
We previously demonstrated that liposomes tagged with the J8 pep-tide, and incorporating DT as a source of T cell 'help' (J8-Lipo-DT) can induce mucosal antibodies and protect mice from a challenge infec-tion when assessed 3-weeks post-vaccination[11]. However, the mechanism and endurance of immunity were not defined. We asked whether we could improve the immunogenicity of the vaccine by further incorporating PHAD (J8-Lipo-DT-PHAD) and sought to define the duration and the mechanism of protection for this vaccine (a schematic is shown in Fig. 1a, b).

BALB/c mice were immunized i.n. on days 0, 21, and 42. In experiment 1 mice were immunized with with J8-Lipo-DT-PHAD, J8-Lipo-D, Lipo-DT-PHAD, Lipo-DT or PBS. Mice immunized with J8-Lipo-DT-PHAD had significantly higher J8-specific salivary IgA and IgG (Fig. 1c) and serum IgA and IgG (Supp. Fig. 2a) in comparison to PBS ($P < 0.0001$) and all other cohorts ($P < 0.0001$). However, the J8-specific serum IgG titers (~100) were significantly lower than those we reported following intramuscular (i.m.) vaccination of mice with J8-DT/Alum or J8-CRM/Alum, where titers of between $10^5$–$10^6$ were observed[39–41]. Two-weeks following the last vaccine boost, mice were challenged via the URT with *S. pyogenes* 2031[11]. Mice were euthanized on day 3 post-infection and NALT and lungs excised to enumerate tissue bacterial burden (Fig. 1d). Both vaccines (J8-Lipo-DT-PHAD or J8-Lipo-DT) resulted in significantly reduced bacterial load in NALT in comparison to PBS ($P < 0.001$ for J8-Lipo-DT-PHAD and $P < 0.01$ for J8-Lipo-DT). Mice vaccinated with J8-Lipo-DT-PHAD had significantly reduced bacterial burden in comparison to mice vaccinated with Lipo-

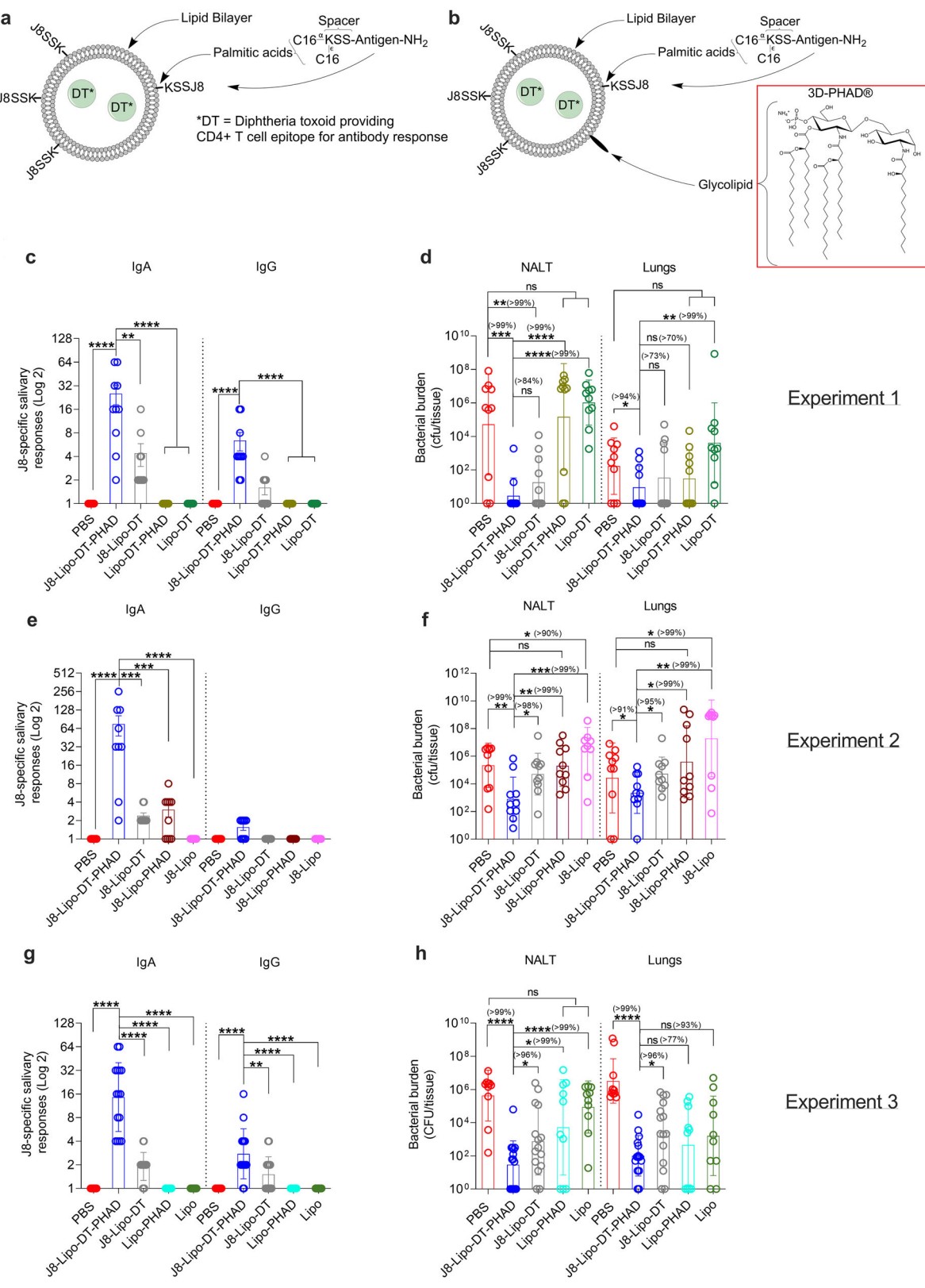

DT-PHAD ($P < 0.001$) or Lipo-DT ($P < 0.0001$). There was no statistical difference in bacterial burdens in the NALT between control mice immunized with PBS and control mice immunized with Lipo-DT or Lipo-DT-PHAD. In the lungs, the bacterial burden was low overall; however, again there was no significant difference in bacterial burdens between PBS, Lipo-DT, and Lipo-DT-PHAD control mice (Fig. 1d). To specifically assess the role of DT or PHAD in the vaccine construct, in

experiment 2 we immunized mice i.n. with J8-Lipo-DT-PHAD ('complete' vaccine), J8-Lipo-DT (without PHAD), J8-Lipo-PHAD (without DT), J8-Lipo (without DT and PHAD) or PBS. As previously observed, serum IgA and IgG responses were lower than reported following i.m. vaccination (Supp. Fig. 2b). Mice immunized with J8-Lipo-DT-PHAD had significantly higher J8-specific salivary IgA in comparison to PBS ($P < 0.0001$), J8-Lipo-DT, J8-Lipo-PHAD ($P < 0.001$) and J8-Lipo

**Fig. 1 | Idealized schematic and immunogenicity and efficacy of J8-Lipo-DT and J8-Lipo-DT-PHAD.** Diphtheria toxoid (DT) was encapsulated within neutral lipids. J8-KSS-(C16)$_2$ and PHAD were inserted into the liposome membrane, both internally and externally. For simplicity, J8-KSS-(C16)$_2$ and PHAD are displayed as being external to the liposome. Idealized schematics of J8-Lipo-DT (without PHAD) (**a**) and J8-Lipo-DT-PHAD (**b**) are shown. **c–h** Immunogenicity and protective efficacy of J8-Lipo-DT and J8-Lipo-DT-PHAD in BALB/c mice. Three experiments were conducted to assess the role of vaccine components in protection. In the first experiment, BALB/c mice ($n = 10$/group; female, 4–6 weeks) were immunized i.n. with PBS, J8-Lipo-DT-PHAD, J8-Lipo-DT, Lipo-DT-PHAD, or Lipo-DT on days 0, 21, and 42 (**c**, **d**). In the second experiment, BALB/c mice were immunized i.n. with PBS, J8-Lipo-DT-PHAD, J8-Lipo-DT, J8-Lipo-PHAD, or J8-Lipo on days 0, 21, and 42 (**e**, **f**). In the third experiment, BALB/c mice were immunized i.n. with PBS, J8-Lipo-DT-PHAD, J8-Lipo-DT, Lipo-PHAD, or Lipo on days 0, 21, and 42 (**g**, **h**). One-week post-last-boost J8-specific salivary IgA and IgG (**c**, **e**, **g**) responses were measured by ELISA and are represented as mean ± SEM. Two weeks post-last-boost, the mice were infected via the URT with *S. pyogenes* 2031 ($5 \times 10^6$ cfu/mouse). On day 3 post-infection all surviving mice were euthanized and NALT and lungs were collected to determine *S. pyogenes* burden (**d**, **f**, **h**), data are represented as the geomean ± geometric SD (cfu/tissue) on a Log 10 scale. Statistical analysis was performed using a nonparametric, unpaired Mann–Whitney *U* test on the raw values of the test groups (one-tailed−ns $p > 0.05$; *$p \le 0.05$; **$p < 0.01$; ***$p < 0.001$; ****$p < 0.0001$). The percent reduction in bacterial burden was calculated by comparing the geomean (cfu/tissue) of the group of interest with the geomean (cfu/tissue) of the test group. Experiments were repeated once to confirm the results. Source data are provided as a Source Data file.

($P < 0.0001$) (Fig. 1e). Following URT challenge with *S. pyogenes* 2031, J8-Lipo-DT-PHAD immunized mice showed significantly better protection ($P < 0.05$–0.001) in comparison to all other cohorts in the NALT and lungs (Fig. 1f), thus highlighting that both DT and PHAD are critical to vaccine immunogenicity and efficacy. Finally, to clearly demonstrate the role of vaccine antigens, we conducted a third immunogenicity and efficacy study (experiment 3) which included the 'carrier/delivery system' control groups Lipo-PHAD and Lipo (Fig. 1g, h). This experiment also included the vaccine groups J8-Lipo-DT-PHAD, J8-Lipo-DT, and the negative control PBS. In concordance with the last two experiments, mice immunized with J8-Lipo-DT-PHAD had significantly higher antigen-specific salivary IgA responses in comparison to all other groups tested (Fig. 1g). Following challenge, mice immunized with J8-Lipo-DT-PHAD had significantly reduced bacterial burden in the NALT in comparison to PBS ($p < 0.0001$), J8-Lipo-DT ($p < 0.05$), Lipo-PHAD ($p < 0.05$) and Lipo ($p < 0.0001$) vaccinated cohorts (Fig. 1h). J8-Lipo-DT-PHAD immunized mice also had a significant reduction in lung bacterial burden in comparison to PBS ($p < 0.0001$) and J8-Lipo-DT ($p < 0.05$). In addition, the J8-Lipo-DT-PHAD cohort had 77% and 93% reduction in lung bacterial burden in comparison to Lipo-PHAD or Lipo respectively, however, this did not reach significance. Serum IgA and IgG levels were comparable to those observed in the previous two experiments (Supp. Fig. 2c). Since there was no significant difference between the control liposomes and PBS for all three experiments, for all further experiments involving a challenge, PBS mice were included as the control.

We also graphed the bacterial burden data from 11 different J8-Lipo-DT-PHAD vaccination-challenge experiments and 4 different J8-Lipo-DT vaccination-challenge experiments. For all 11 experiments involving a total of 115 mice vaccinated with J8-Lipo-DT-PHAD and 110 control mice, there was 88% and 93% mean cumulative reduction in comparison to PBS in the NALT and lungs, respectively ($P < 0.0001$; Supp. Fig. 3a–d). For the 4 experiments involving J8-Lipo-DT, similar reductions in bioburden in comparison to PBS were seen in NALT and lungs as recorded for the J8-Lipo-DT-PHAD experiments with overall mean reductions of 97% and 89%, respectively ($p < 0.01$–0.0001; Supp. Fig. 3e–h).

### J8-Lipo-DT-PHAD immunization induces antigen-specific Th17 cells in the lungs

Th17 cells are critical immune elements against mucosal *S. pyogenes* infections. They secrete IL-17 which has been shown to play a protective role in host defence against *S. pyogenes* infections[17,42–44]. Using intracellular cytokine staining (ICS; Fig. 2a) we observed that lung cells from mice immunized i.n. with J8-Lipo-DT-PHAD, six weeks previously, had significantly more antigen (J8-DT)-specific IL-17A$^+$CD4$^+$ (Th17) cells in comparison to mice immunized with PBS, Lipo-DT, Lipo-DT-PHAD, Lipo-PHAD, J8-Lipo-PHAD or Lipo (Fig. 2b). In comparison to the J8-Lipo-DT and J8-Lipo vaccinated groups there was >3.5-fold increase in Th17 cells, however, this did not reach significance (Fig. 2b). We also measured IL-17A in cell culture supernatants from vaccinated lung

(Fig. 2c) and spleen (Fig. 2d) cells stimulated ex-vivo with J8-DT and the individual antigenic components, J8 and DT. Stimulation of J8-Lipo-DT-PHAD vaccinated lung cells with J8-DT, J8 or DT induced a significant response in comparison to PBS and other control groups (Fig. 2c). In the spleen, there was also a significant response to J8-DT, but the level of the response was much reduced in comparison to the lung (Fig. 2c). However, spleen cells stimulated ex-vivo with J8 or DT alone did not induce a detectable IL-17A response (Fig. 2d), suggesting that these cells may have been recruited to the lungs following airway exposure to the cognate antigen[43]. We also noted that Lipo-DT-PHAD vaccinated mice also responded to J8 (Fig. 2c), which may be due to the known sequence similarities between J8 and DT[7].

### J8-Lipo-DT-PHAD immunization leads to enduring immunity and protection

The longevity of an immune response and the ability of infection to rapidly recall an immune response are critical parameters of a successful vaccine. To assess this, BALB/c mice were immunized with J8-Lipo-DT, J8-Lipo-DT-PHAD, or PBS. The experiment was divided into two cohorts−cohort 1 was designated for *S. pyogenes* URT challenge and cohort 2 was designated for measurement of antibody levels and secreted cytokines (Fig. 3a; schematic of experimental protocol). Longevity of vaccine protection was assessed 30 weeks post-vaccination (day-252). Antibody levels were measured in the saliva (Fig. 3b) and serum (Fig. 3c) two days before *S. pyogenes* URT challenge. The experiment was terminated on day 1 post-challenge due to 60% of control mice presenting with severe clinical signs for humane end-point (Fig. 3d). Mice immunized with J8-Lipo-DT-PHAD had significantly lower clinical scores (Fig. 3d), coinciding with significantly reduced *S. pyogenes* in their NALT and lungs compared to the PBS control group (98% and 99% reduction, respectively; $P < 0.05$ for both) (Fig. 3e). Clinical scores are enumerated between 1–3 for ruffled fur, hunched posture, response to stimuli, level of consciousness/activity, respiration quality, eyes with excess secretions, blood in throat swabs and >15% weight loss in comparison pre-challenge. A score above 3 for any of these categories or a total score of above 14 instigates immediate euthanasia[45]. Mice are scored with observer blinded to the treatment. J8-Lipo-DT-immunized mice also had reduced bacterial counts compared to PBS controls, however not statistically significant (Fig. 3e). Mice immunized with J8-Lipo-DT-PHAD had reduced bacterial counts compared to mice immunized with J8-Lipo-DT in NALT and lungs (58% and 99% reduction, respectively, although not statistically significant) (Fig. 3e).

Antibody levels in serum and saliva were monitored for <1-year (454 days), at which time they were i.n. boosted with $10^4$ cfu (low dose) of *S. pyogenes* or PBS (day 455) to determine whether there was evidence of anamnestic (rapid) immune responses, which would indicate functional immunological memory. Vaccine-specific antibody and cytokine responses (as measures of humoral and cellular immunity, respectively) were determined before and after challenge. At all times up to day 454, J8-Lipo-DT-PHAD-immunized mice had significantly

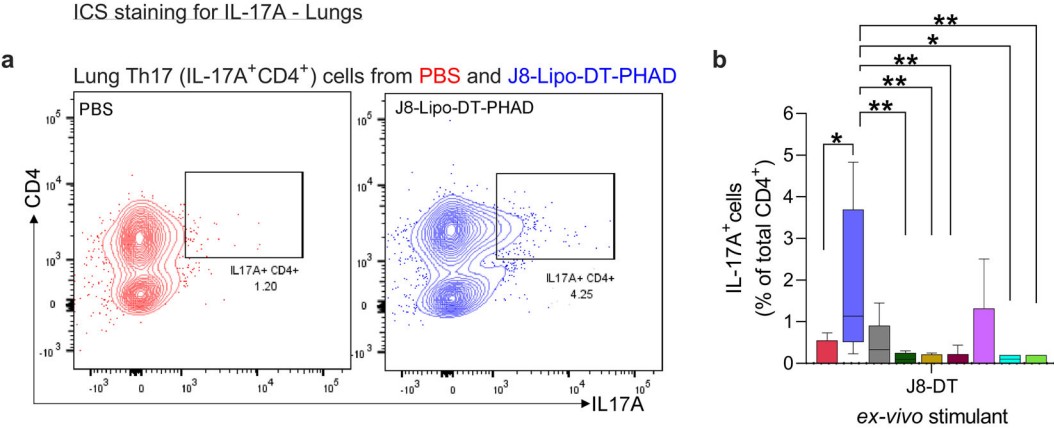

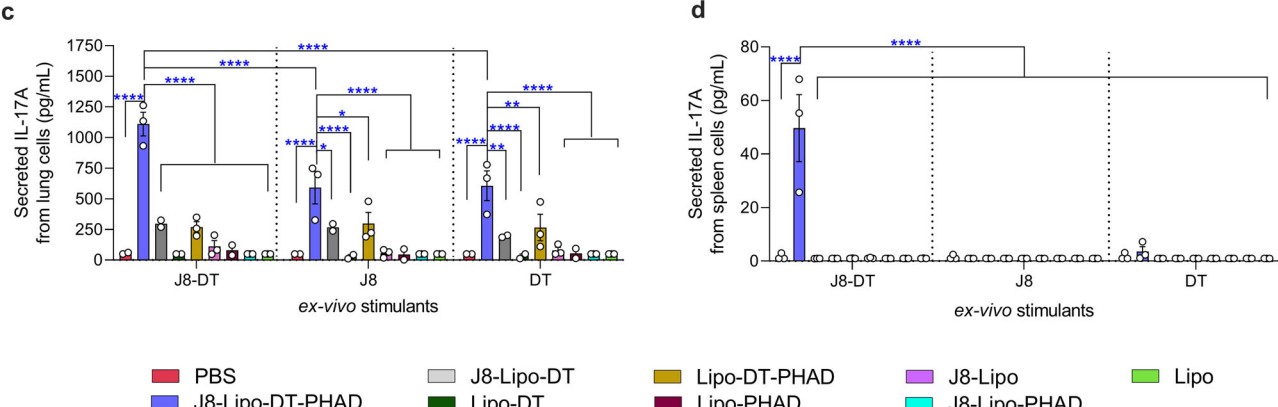

**Fig. 2 | J8-Lipo-DT-PHAD-induced Th17 cells.** BALB/c mice ($n = 5$ mice/group; female, 4–6 weeks old) were immunized i.n. on days 0, 21, and 42 with PBS, J8-DT-PHAD, J8-Lipo-DT, Lipo-DT, Lipo-DT-PHAD, J8-Lipo-PHAD, J8-Lipo, Lipo-PHAD or Lipo. Six weeks post-last-immunization mice were euthanized, and lungs excised. Lung cells were stimulated with J8-DT (0.05 mg/well) or media for 16 h and IL-17A⁺ CD4⁺ cells were quantified using intracellular cytokine staining (ICS). Flow cytometry contour plot representing the gating strategy used to identify IL-17-A⁺CD4⁺ (Th17) cells from PBS and J8-Lipo-DT-PHAD immunized mice (**a**). A boxplot (min/max; medium with SEM) representing the frequency of IL-17A⁺ from CD4⁺ cells (**b**). Data are normalized to the media stimulation control (% IL-17A⁺ cells from the J8-DT stimulated group minus % IL-17A⁺ cells from the media stimulated group [to remove background]). Statistical analysis was performed using a nonparametric, unpaired

Mann–Whitney $U$ test (one-tailed) (*$p < 0.05$, **$p < 0.01$) to compare the different immunization cohorts to J8-Lipo-DT-PHAD. **c**, **d** Secreted IL-17A from immune cells isolated from lung digests following intranasal vaccination and ex-vivo stimulation. BALB/c mice ($n = 3$/group; female, 4–6 weeks old) were immunized as above. Lung (**c**) and spleen (**d**) cells were stimulated ex-vivo with J8-DT (0.05 mg/well) or media. At 72 h post-stimulation, supernatants were isolated and levels of secreted IL-17A were assessed in culture supernatants using a cytometric bead array. Data are normalized to the media stimulation control and represented as mean ± SEM. Statistical analysis was performed using two-way ANOVA, corrected for multiple comparisons using Sidak's multiple comparison test (*$p < 0.05$; **$p < 0.01$; ***$p < 0.001$; ****$p < 0.0001$). Experiments were repeated once to confirm the results. Source data are provided as a Source Data file.

higher antibody titers in comparison to J8-Lipo-DT-immunized mice. This occurred with J8-specific salivary IgA ($P < 0.0001$; Fig. 3f), serum IgA ($P < 0.0001$; Fig. 3g), salivary IgG ($P < 0.0001$; Fig. 3h), and serum IgG ($P < 0.0001$; Fig. 3i). However, by day 454, all antibody titers had dropped significantly. Nevertheless, following a boost on day 455 with *S. pyogenes* (Fig. 3f–i), a rapid antibody anamnestic response was observed in both serum and salivary compartments in mice immunized with either J8-Lipo-DT or J8-Lipo-DT-PHAD in comparison to the PBS group, indicating the presence of long-lived memory B-cells in all mice. Antibodies were assessed six days post i.n. boost with *S. pyogenes*. Following boosting, the titers were significantly higher in mice that had been vaccinated with J8-Lipo-DT-PHAD in comparison to J8-Lipo-DT (Fig. 3f–i).

To investigate long-term cellular mediators of vaccine-mediated immunity, secreted cytokines (IL-17A, IL-6, and IL-2) were assessed in mice immunized with J8-Lipo-DT-PHAD, J8-Lipo-DT, or PBS on day 50 (one-week post-last immunization), and days 455 and 461 (pre-and 6-days post in-vivo boosting with *S. pyogenes*, respectively). Cytokines were measured in supernatants from spleen cells stimulated ex-vivo

with J8-DT (Fig. 3j–l), J8 or DT (Supp. Fig. 4a–c). IL-17A (Fig. 3j), IL-6 (Fig. 3k) and IL-2 (Fig. 3l) secretion in J8-Lipo-DT-PHAD vaccinated mice on day 50 stimulated ex-vivo with J8-DT were significantly higher than the PBS control mice ($P < 0.05$–0.01). All cytokine responses in the J8-Lipo-DT-PHAD mice had diminished by day 455 but, IL-17A and IL-2 (Fig. 3j, l, respectively) responses were recalled in these vaccinated mice following i.n. boosting with *S. pyogenes* ($P < 0.05$). IL-6 responses were not recalled. Spleen cells from J8-Lipo-DT-immunized mice failed to produce a cytokine response at either day 50 or following boosting.

## J8-Lipo-DT-PHAD-mediated protection is independent of antibody

The data showed that protection induced by the PHAD-containing vaccine is long-lived and is associated with the presence of vaccine-specific antibody and cytokine responses and that both can be boosted by pathogen exposure more than 1-year post-vaccination. To ask whether vaccine-induced antibodies were functional, we pre-incubated *S. pyogenes* 2031 with neat J8-Lipo-DT-PHAD, J8-Lipo-DT, or PBS antisera. Sera were collected three months post-last-vaccine

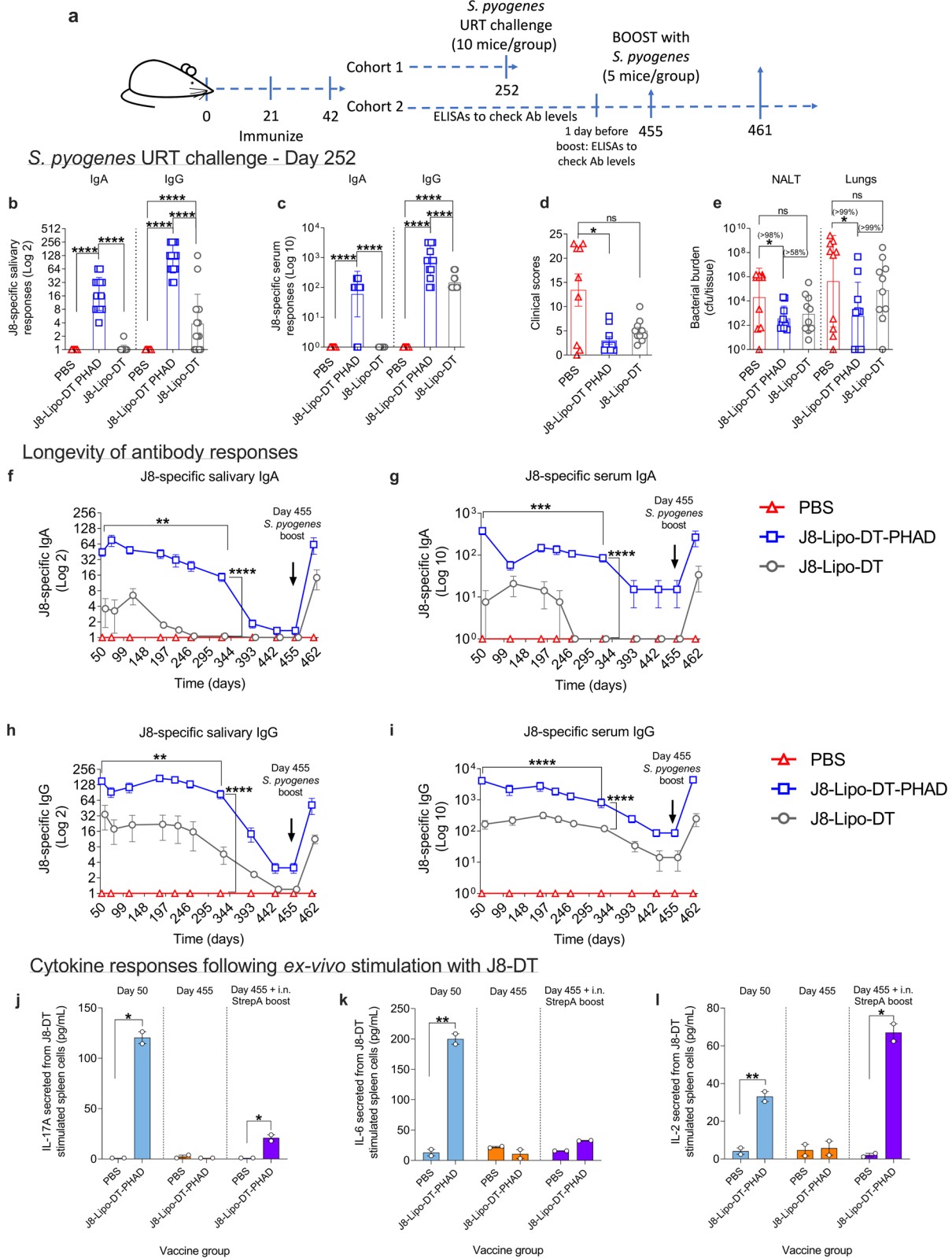

boost and IgG and IgA antibody levels were confirmed (Fig. 4a). Bacterial cells ($5 \times 10^6$ cfu/mouse) in serum were then delivered i.n. to naive BALB/c mice. On day 3 post-infection all surviving mice were euthanized and NALT and lungs were collected to determine *S. pyogenes* burden. There was a significant reduction in bacterial burden in NALT (>99% reduction; $P < 0.0001$) and lungs (>99% reduction; $P < 0.0001$) following pre-incubation with J8-Lipo-DT-PHAD

antiserum in comparison to the PBS antiserum cohort (Fig. 4b). The reduction in NALT for the J8-Lipo-DT-PHAD cohort was also evident in comparison to the cohort that received J8-Lipo-DT antiserum pre-incubated *S. pyogenes* (99% reduction; $P < 0.0001$) and lungs (95% reduction, however not significant) (Fig. 4b). The data indicated that vaccination with J8-Lipo-DT-PHAD gave rise to functional antibodies. Given the enhanced protection seen with J8-Lipo-DT-PHAD and the

**Fig. 3 | Longevity and protective efficacy of immune responses following i.n. immunization with J8-Lipo-DT-PHAD and J8-Lipo-DT.** BALB/c mice (*n* = 15/group, female, 4–6 weeks old) were immunized i.n. with PBS, J8-Lipo-DT-PHAD or J8-Lipo-DT. Schematic of experimental protocol (**a**). BALB/c mice (*n* = 10/group; female) were rested for >30 weeks and J8-specific antibody titers were measured in saliva (**b**) and serum samples (**c**), data are represented as mean ± SEM. Mice were infected via the URT with *S. pyogenes* 2031 (5 × 10⁶ cfu/mouse). On day 1 post-infection, mice were monitored for signs of distress/illness as per the approved score sheet. Sixty percent of control (PBS immunized) mice showed severe clinical symptoms on day 1–leading to premature experimental termination), data are represented as mean ± SEM (**d**). NALT and lungs were collected to determine *S. pyogenes* burden, data represent the geomean ± geometric SD (cfu/tissue) on a Log 10 scale (**e**). In a second cohort of mice (*n* = 5/group, female), antibody titers in saliva and serum were followed for 454 days. On day 455, mice were boosted with *S. pyogenes* 2031. Saliva and serum samples were collected on day 461 and antibody responses were measured by ELISA (**f–i**). J8-specific salivary IgA (**f**), J8-specific

serum IgA (**g**), J8-specific salivary IgG (**h**), and J8-specific serum IgG (**i**) are represented as mean ± SEM. Statistical analysis was performed using a nonparametric, unpaired Mann–Whitney *U* test (one-tailed) on the raw CFU values of the test groups (ns; *$p \leq 0.05$; **$p < 0.01$; ***$p < 0.001$; ****$p < 0.0001$). The percent reduction in bacterial burden was calculated by comparing the geomean (cfu/tissue) of the group of interest with the geomean (cfu/tissue) of the test group. **j–l** J8-Lipo-DT-PHAD-induced cytokine secretion. Spleens were excised and splenocytes were stimulated ex-vivo with J8-DT (0.05 mg/well) or media for 72 h and supernatants were collected to measure IL-17A (**f**), IL-6 (**g**), and IL-2 (**h**) in J8-Lipo-DT-PHAD or PBS vaccinated mice (*n* = 2/group/time-point, female). Cytokines were measured on day 50 (1-week post-last vaccine boost), day 455 (following rest), and following i.n. in-vivo stimulation with *S. pyogenes* 2031 (day 455 + i.n. StrepA boost). Data are represented mean ± SEM normalized to media stimulation control. Statistical analysis was performed using an unpaired *t* test with Welch's correction (*$p \leq 0.05$; **$p < 0.01$). Source data are provided as a Source Data file.

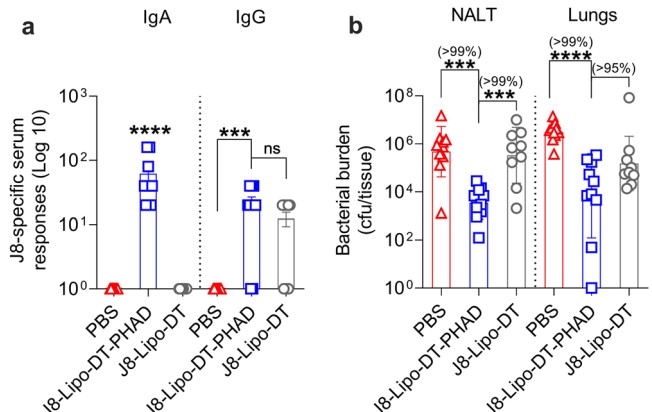

**Fig. 4 | Preincubation challenge assay to assess functionality of vaccine-induced antibodies.** *S. pyogenes* 2031 (5 × 10⁶ cfu/mouse) were incubated in-vitro with neat serum from PBS, J8-Lipo-DT-PHAD or J8-Lipo-DT-immunized BALB/c mice. IgG and IgA responses in donor mice are shown (**a**), data are represented as mean ± SEM. Following incubation, bacterial cells were delivered i.n. to naive BALB/c mice (10 µL/mouse; *n* = 10 mice/group, female). On day 3 post-infection all surviving mice were euthanized and NALT and lungs were collected to determine *S. pyogenes* burden (**b**), data represent the geomean ± geometric SD (cfu/tissue) on a Log 10 scale. Statistical analysis was performed using a nonparametric, unpaired Mann–Whitney *U* test (one-tailed) on the raw CFU values of the test groups (***$p < 0.001$; ****$p < 0.0001$). The percent reduction in bacterial burden was calculated by comparing the geomean (cfu/tissue) of the group of interest with the geomean (cfu/tissue) of the test group. The experiment was repeated once to confirm the results. Source data are provided as a Source Data file.

lack of a cellular memory response following vaccination with J8-Lipo-DT, we pursued further investigation with the J8-Lipo-DT-PHAD vaccine candidate.

The data demonstrated that functional antibodies were induced post-vaccination, however they did not explicitly show that the antibodies were essential to immunity. We asked whether J8-Lipo-DT-PHAD vaccine-induced immunity could occur in the absence of antibody, and studied protection in pIgR⁻/⁻ mice (which do not have mucosal antibody responses)[46] and µMT mice (which lack mature B-cells, membrane-bound IgM and serum and salivary antibodies)[47]. Vaccinated pIgR⁻/⁻ mice did not have detectable salivary antibodies but had J8-specific serum IgA and IgG antibodies (Fig. 5a). Vaccinated and control mice were challenged with *S. pyogenes* at 5 × 10⁶ cfu/mouse via the URT 2 weeks post-vaccination. On day 3 post-challenge, NALT and lungs were collected. Vaccination resulted in significantly reduced bacterial burden in the NALT and in the lungs in comparison to the control cohort (Fig. 5b; 99% reduction, *P* < 0.01). Thus, although vaccinated normal mice contain functional antibodies, the data suggested

that neither mucosal IgA nor IgG were required for protection. To determine whether serum IgG may be required, we vaccinated µMT mice. One-week following vaccination neither mucosal nor serum J8-specific IgG or IgA were detected in the µMT mice, but control mice responded as previously. Two weeks following vaccination, mice were challenged via the URT and we again observed a significant reduction in the bacterial burdens in the NALT (99%; *P* < 0.001) and lungs (99%; *P* < 0.01) of the vaccinated mice in comparison to the control cohort (Fig. 5c). There was no significant difference between the vaccinated knock-out (KO) mice strains (pIgR⁻/⁻ or µMT) or the vaccinated wild-type (WT) mice (C57BL/6). However, KO mice may have other compensatory mechanisms that may contribute to protection, thus making it difficult to completely exclude the role of antibodies in this experimental model.

We asked whether vaccine-induced protection seen in the URT of mice lacking mucosal antibodies could be explained by an enhanced cellular response in these mice. pIgR⁻/⁻ and C57BL/6 mice were vaccinated with J8-Lipo-DT-PHAD and 2 weeks following vaccination, spleens were excised, and cells were stimulated with J8-DT or media. Mice lacking mucosal antibodies had a significantly enhanced IL-17A response to J8-DT compared to mice with mucosal antibodies, whereas the IFN-γ response was significantly higher in mice with mucosal antibodies (Fig. 5d–f). IL-17A has been shown to contribute to the control of *S. pyogenes* i.n. infection[17,48]. The IL-2 response was similar in both groups. An inverse relationship between IFN-γ and IL-17A has been previously reported in other systems[49], but not in mice lacking mucosal antibodies. Given the importance of both mucosal antibodies and IL-17A in defence against extracellular mucosal pathogens[17,42–44], it may not be surprising that IL-17A is higher following intranasal vaccination of pIgR⁻/⁻ mice. IL-17 is known to inhibit IFN-γ expression by targeting TBet[50], which may provide the explanation for the reduced IFN-γ expression in pIgR⁻/⁻ mice.

In keeping with previous data in other systems[17,51], our data suggested an important role for IL-17A in liposomal vaccine-induced mucosal protection. To confirm that role, we vaccinated *Il17*⁻/⁻ mice with J8-Lipo-DT-PHAD or PBS. We observed that *Il17*⁻/⁻ mice had mucosal and serum antibody levels comparable to previous experiments (Fig. 6a, b). However, *Il17*⁻/⁻ mice vaccinated with J8-Lipo-DT-PHAD were not protected following URT challenge with *S. pyogenes* 2031 (Fig. 6c). These data demonstrated that cellular immunity, and IL-17 in particular, are key contributors to the liposomal vaccine-mediated protection against *S. pyogenes* 2031 infection.

## J8-Lipo-DT-PHAD-mediated protection is lost following CD4⁺ Tcell depletion

To more definitively assess the role of T cells in protection, CD4⁺ T cell depletion studies were performed. BALB/c mice were immunized with J8-Lipo-DT-PHAD or PBS on days 0, 21, and 42. On day 50, serum and

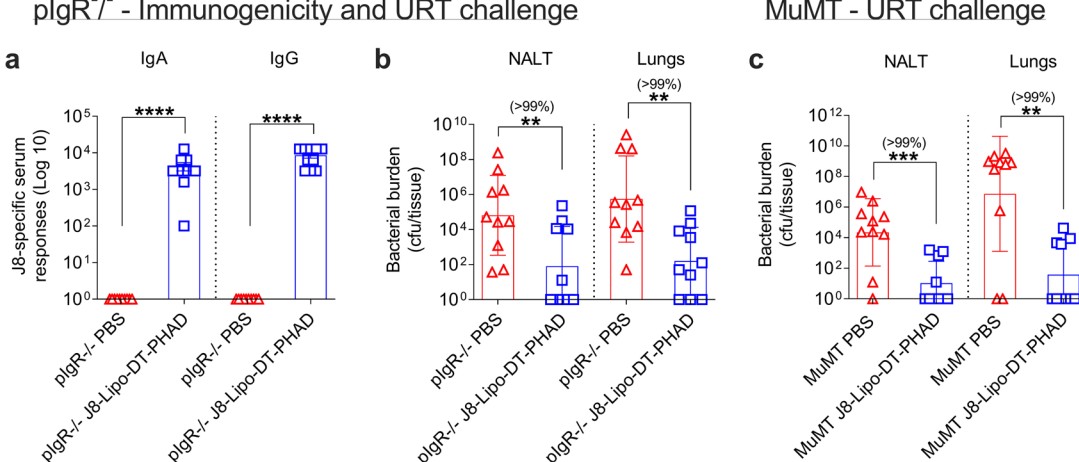

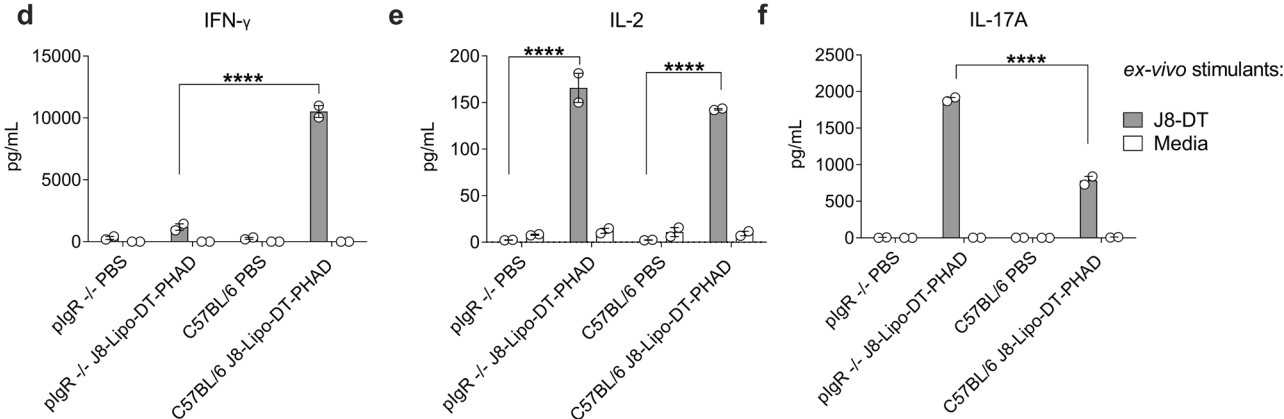

**Fig. 5 | Role of humoral responses in J8-Lipo-DT-PHAD-mediated protection.**
**a**, **b** Role of mucosal antibodies. pIgR$^{-/-}$ mice ($n = 10$/group; male and female, 4–6 weeks old) were immunized i.n. with J8-Lipo-DT-PHAD or PBS. One-week post-last boost serum samples were analyzed for J8-specific serum IgA and IgG antibody responses by ELISA (**a**), represented as mean ± SEM. Two weeks after last boost, the mice were infected via the URT with *S. pyogenes* 2031 ($5 × 10^6$ cfu/mouse). On day 3 post-infection all surviving mice were euthanized and NALT and lungs collected to determine *S. pyogenes* burden (**b**), data represent the geomean ± geometric SD (cfu/tissue) on a Log 10 scale. **c** Role of serum antibodies. μMT (MuMT) mice ($n = 10$/group; male and female, 4–6 weeks old) were immunized i.n. with J8-Lipo-DT-PHAD or PBS. Two weeks after last boost, the mice were infected via the URT with *S. pyogenes* 2031 ($5 × 10^6$ cfu/mouse). On day 3 post-infection all surviving mice were euthanized and NALT and lungs collected to determine *S. pyogenes* burden, data represented as the geomean ± geometric SD (cfu/tissue) on a Log 10 scale. Statistical analysis was performed using a nonparametric, unpaired Mann–Whitney

*U* test on the raw CFU values (one-tailed) for both pIgR$^{-/-}$ and μMT experiments (**$p < 0.01$; ****$p < 0.0001$). The percent reduction in bacterial burden was calculated by comparing the geomean (cfu/tissue) of the PBS group with the geomean (cfu/tissue) of the vaccine group. **d–f** Cytokine expression in pIgR$^{-/-}$ mice vaccinated intranasally with J8-Lipo-DT-PHAD. pIgR$^{-/-}$ and C57BL/6 mice ($n = 3$/group; male/female, 4–6 weeks old) were immunized i.n. with J8-Lipo-DT-PHAD or PBS. Two weeks after last boost, mice were euthanized, and spleens were excised. The splenocytes were stimulated ex-vivo either with J8-DT (50 μg/well) or media (left unstimulated). At 72 h post-stimulation, supernatants were isolated and levels of (**d**) IFN-γ, (**e**) IL-2, and (**f**) IL-17A were assessed using cytometric bead array as per manufacturer's instructions. Data are represented as mean ± SEM. Statistical analysis was performed using two-way ANOVA, corrected for multiple comparisons using Sidak's multiple comparison test (****$p < 0.0001$). All experiments were repeated once to confirm the results. Source data are provided as a Source Data file.

saliva samples were collected, and antibody titers were assessed. J8-specific salivary IgA titers ranged from 8–128 and J8-specific salivary IgG titers ranged from 0–64 (Fig. 6d). Serum IgA and IgG responses were similar to those shown previously (Fig. 6e). Prior to challenge, half of the immunized and control cohorts ($n = 10$/group) were depleted of CD4$^+$ T cells by intraperitoneal administration of a rat anti-mouse CD4$^+$ mAb (Supp. Figs. 5–6). Control animals received non-specific purified rat IgG. Following URT challenge, immunized-depleted mice had significantly increased ($P < 0.01$) bacterial burden in NALT and lungs when compared to immunized-non-depleted mice (Fig. 6f). Furthermore, the data showed that all the protection afforded by the J8-Lipo-DT-PHAD vaccinated wild-type (non-depleted) mice was lost by removal of CD4$^+$ T cells at the time challenge.

## J8-Lipo-DT-PHAD immunization induces tissue-resident memory (T$_{RM}$) cells in the lungs

T$_{RM}$, a specific subset of memory T cells resident in both the upper and lower respiratory tract, have been shown to play an instrumental role in local immune responses against bacterial infections[52,53]. We dissected the CD4$^+$ T$_{RM}$ response in the lungs of mice immunized with, J8-Lipo-DT-PHAD, J8-Lipo-DT, Lipo-DT, Lipo-DT-PHAD, J8-Lipo-PHAD, J8-Lipo, Lipo-PHAD, Lipo or PBS at 6 weeks post-last vaccine boost (Fig. 7a, b). CD4$^+$ T$_{RM}$ (CD103$^-$CD69$^+$) cells were identified from within the pool of effector memory cells (EM; CD62L$^-$CD44$^+$) which circulate through non-lymphoid organs and provide accelerated clearance of pathogen. CD69 is a key marker that distinguishes memory T cells in tissues from those that are in

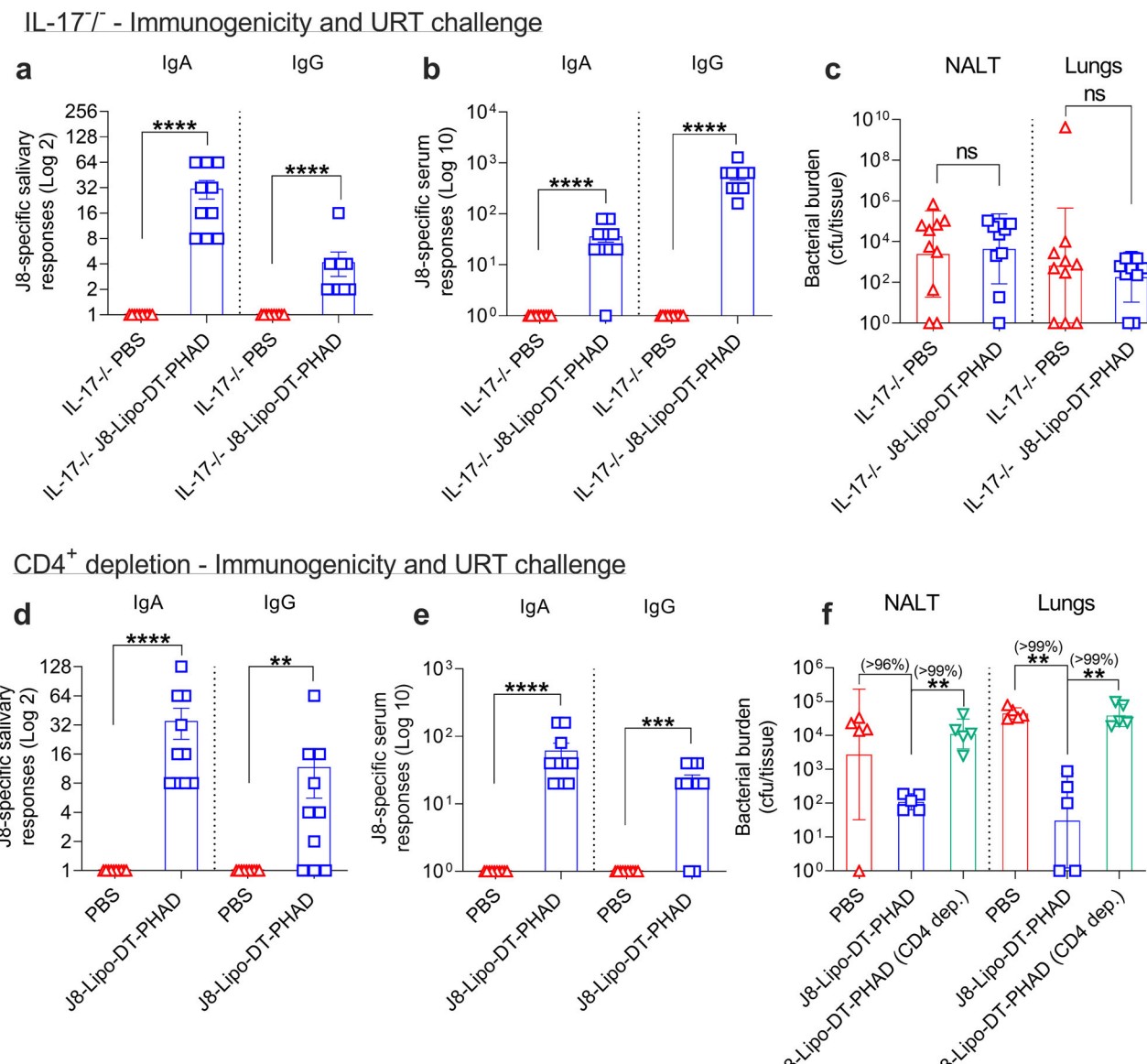

**Fig. 6 | Role of IL-17 and CD4⁺ T cells in J8-Lipo-DT-PHAD-mediated protection.**
**a–c** Role of IL-17. *Il17⁻/⁻* mice (n = 10/group; female, 4–6 weeks old) were immunized i.n. with J8-Lipo-DT-PHAD or PBS. One-week post-last-boost J8-specific salivary IgA and IgG (**a**) and J8-specific serum IgA and IgG (**b**) responses were measured by ELISA and are represented as mean ± SEM. Two weeks after last vaccine boost, the mice were challenged via the URT with *S. pyogenes* 2031 (5 × 10⁶ cfu/mouse). On day 2 post-challenge, all surviving mice were euthanized and NALT and lungs collected to determine *S. pyogenes* burden (**c**), data represent the geomean ± geometric SD (cfu/tissue) on a Log 10 scale. **d–f** Role of CD4⁺ T cells. BALB/c mice (n = 10/group; female, 4–6 weeks old) were immunized i.n. with J8-Lipo-DT-PHAD or PBS. One-week post-last-vaccine boost saliva and serum samples were analyzed to measure J8-specific salivary IgA and IgG (**d**) and serum IgA and IgG (**e**) antibody responses by ELISA and are represented as mean ± SEM. Two-weeks after last boost, half of the mice were administered anti-mouse CD4⁺ mAb (GK1.5) intraperitoneally (i.p.) at an

optimized dose of 215 mg/mouse on days −4 and −1 before challenge (day 0) to achieve >99% depletion of CD4⁺ cells in all tissues tested. The other half of the mice were administered non-specific rat IgG (non-depleted cohort). Mice were challenged via the URT with *S. pyogenes* 2031 (5 × 10⁶ cfu/mouse). On day 3 post-infection, all surviving mice were euthanized and NALT and lungs collected to determine *S. pyogenes* burden (**f**). The CD4⁺ depletion experiment was repeated once to confirm the results. Data are represented as geomean ± geometric SD (cfu/tissue) on a Log 10 scale. Statistical analysis was performed using a nonparametric, unpaired Mann–Whitney *U* test on the raw CFU values (one-tailed) to compare against the PBS group (ns *p* > 0.05; **p* < 0.01; ***p* > 0.001 ****p* < 0.0001) or J8-Lipo-DT-PHAD vaccinated group (***p* < 0.01). The percent reduction in bacterial burden was calculated by comparing the geomean (cfu/tissue) of the group of interest with the geomean (cfu/tissue) of the test group. Source data are provided as a Source Data file.

circulation while CD103 is expressed predominantly by CD8⁺ T$_{RM}$ cells[54]. Supp. Fig. 7 represents a flow cytometry contour plot of lung T$_{RM}$ cells from PBS and J8-Lipo-DT-PHAD vaccinated mice. CD103⁻CD69⁺ CD4⁺ T$_{RM}$ cells specifically accumulated in the lungs of mice vaccinated with J8-Lipo-DT-PHAD in comparison with PBS, but not in lungs of mice immunized with the other vaccine formulations (Fig. 7). Importantly, there were significantly more T$_{RM}$

cells in the J8-Lipo-DT-PHAD vaccinated group in comparison to the J8-Lipo-DT (*P* < 0.01) vaccinated group, providing evidence that addition of PHAD in the vaccine formulation is critical for induction of CD103⁻CD69⁺ CD4⁺ T$_{RM}$ in the lungs. There was no significant difference in CD103⁺CD69⁺ or CD103±CD69⁺ (total CD69⁺) CD4⁺ T$_{RM}$ cells between the J8-Lipo-DT-PHAD vaccinated cohort and the PBS cohort in the lungs.

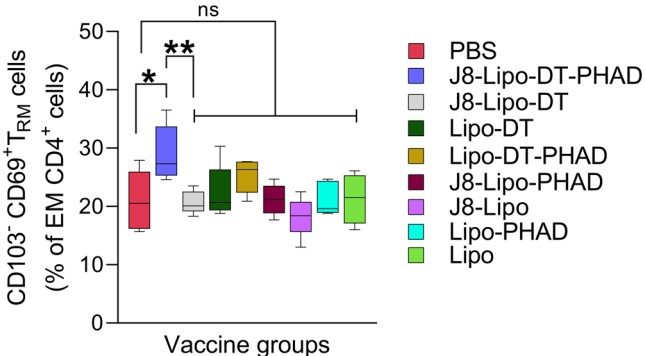

**Fig. 7 | Tissue-resident memory (T$_{RM}$) cells in the lungs of vaccinated mice.** BALB/c mice ($n$ = 5 mice/group; female, 4–6 weeks old) were immunized i.n. on days 0, 21, and 42 with PBS, J8-Lipo-DT-PHAD, J8-Lipo-DT, Lipo-DT, Lipo-DT-PHAD, J8-Lipo-PHAD, J8-Lipo, Lipo-PHAD or Lipo, and lungs excised. Data represented as the frequency of T$_{RM}$ (CD103$^-$CD69$^+$) cells from effector memory (EM; CD62L$^-$CD44$^+$) CD4$^+$ cells (boxplot−min/max; medium with SEM). Statistical analysis was performed using one-way ANOVA (*$p$ < 0.05; **$p$ < 0.01) for multiple comparisons (Dunnett's multiple comparisons test). Samples were run on a BD LSRFortessa cytometer and data analyzed with FlowJo v10.8 software (BD Biosciences). The experiment was repeated once to confirm the results. Source data are provided as a Source Data file.

## Macrophages and neutrophil are the key contributors to J8-Lipo-DT-PHAD-mediated protection

While CD4$^+$ T cells are known to provide help for ongoing antibody responses following challenge[55], the absolute loss of protection that we observed following depletion, together with the inability of the vaccine to protect *Il17*$^{-/-}$ mice (which had normal antibody responses), suggested that cellular immunity was playing a major role in protection. Since CD4$^+$ T cells cannot directly kill streptococcus, we asked whether macrophages[21,22] or neutrophils[17], which can be activated by CD4$^+$ T cells, might be the final mediators of immunity. Macrophages and neutrophils are known to contribute to protection against *S. pyogenes*[8,17–20] and we have previously demonstrated that skin-resident macrophages and neutrophils in conjunction with anti-J8 antibody contribute to the control at the local site of infection[8]. Here, we saw that J8-Lipo-DT-PHAD vaccinated mice have significantly more neutrophils in the NALT ($P$ < 0.01), lungs ($P$ < 0.05), and spleen ($P$ < 0.05) in comparison to PBS cohorts (Supp. Fig. 8a–c). No significant difference in macrophages between the vaccinated and PBS mice was noted.

We thus tested the role of macrophages (MΦD [depleted]), neutrophils (NeuD [depleted]), or both macrophages and neutrophils (DD [double depleted]) in vaccine-mediated mucosal immunity (Fig. 8). Untreated (non-depleted; NonD) mice were included as controls. Neutrophils were depleted post-vaccination and prior to challenge using anti-Ly6G mAb (clone 1A8; BioXcell) and macrophages were depleted using Clodronate Liposomes (ClLip; Liposoma). Macrophage and neutrophil depletion were quantified on day 2, relative to challenge on day 0. We observed >98% depletion in lung and spleen neutrophils (CD45$^+$CD11b$^+$Ly6G$^+$) in the vaccinated and control (PBS) cohorts treated with anti-Ly6G mAb (Supp. Fig. 9a(i), b(i), c and e). Macrophages in the lungs and spleen were characterized as either alveolar (CD45$^+$F4/80$^+$CD11c$^+$; Supp. Fig. 9a(ii)) or splenic macrophages (CD45$^+$F4/80$^+$CD11b$^+$CD24$^-$MHCII$^+$; Supp. Fig. 9b(ii)). Following clodronate liposome treatment there was >40% depletion in alveolar macrophages (Supp. Fig. 9d) and >70% depletion in splenic macrophages (Supp. Fig. 9f). F4/80$^+$ macrophage depletion was also confirmed via histopathological staining and demonstrated comparable results (Supp. Fig. 10). Following i.n. immunization with J8-Lipo-DT-PHAD (pre-macrophage depletion), mice had J8-specific salivary IgA and IgG levels similar to those previously observed (Fig. 8a). Following challenge, vaccinated NonD mice demonstrated significantly less

bacterial burden in comparison to PBS NonD mice in the NALT and lungs ($P$ < 0.001; Fig. 8b). Vaccinated NonD mice showed significantly better protection in comparison to all the vaccinated depleted cohorts in the NALT ($P$ < 0.01) and lungs ($P$ < 0.001; Fig. 8b). Even though clodronate liposome treatment led to incomplete macrophage depletion in the lungs, the protective immunity in the treated mice was severely compromised (Fig. 8b). J8-Lipo-DT-PHAD-vaccinated MΦD, NeuD, and DD mice demonstrated significantly less resistance to infection in the spleen compared with vaccinated non-depleted mice ($P$ < 0.01–0.0001; Supp. Fig. 11). The data thus show that macrophages and neutrophils play an important role in immunity. The data are consistent with a mechanism in which macrophages and neutrophils in the URT are activated by CD4$^+$ T cells via IL-17 and then phagocytose the streptococcal organism. The observation that protection is ablated by depletion of either macrophages or neutrophils suggests that concomitant presence of both cell types are essential for protection. It is possible that macrophages are required to activate neutrophils, as has been suggested in other systems[56,57]. Because the vaccinated mice have functional serum antibodies, albeit low levels, it is likely that the macrophages and neutrophils are clearing the streptococci by both opsonic and non-opsonic phagocytosis[8,21–24].

Finally, to assess whether the depletion of macrophages or neutrophils interferes with the IL-17A response, we measured the IL-17A response in lung cells from vaccinated and PBS mice following depletion. We found that MΦD, NeuD, and DD significantly ($P$ < 0.0001) reduces the IL-17A response (Fig. 8c). This is consistent with previous findings that macrophages drive the differentiation of Th17 cells through IL-23[58] and neutrophils themselves are a potential source of IL-17A[59,60]. IL-17A plays a vital role in protecting the host from infection at mucosal sites including the lungs[61]

## Discussion

The data presented here show that intranasal administration of a liposomal vaccine incorporating the peptide antigen, J8, and the immunostimulatory glycolipid, PHAD, is able to induce both cellular and humoral immune responses. Intranasal boosting with *S. pyogenes* led to anamnestic responses of antibody and cytokines more than 1-year post-vaccination. However, despite inducing a strong IgA antibody response, vaccinated mice deficient in mucosal IgA and IgG (pIgR$^{-/-}$ mice) or total antibodies (µMT mice) were protected following URT challenge with *S. pyogenes*. Vaccinated mice lacking mucosal antibodies had significantly higher secreted IL-17A levels in comparison to vaccinated wild-type controls. Furthermore, vaccinated mice deficient in IL-17A were not protected following *S. pyogenes* URT infection, demonstrating a role of IL-17A which in turn is 'regulated' by macrophages and neutrophils[58,59].

CD4$^+$ T$_{RM}$ cells localize at the site of infection and mount a rapid and targeted immune response against bacterial infections[62]. They serve as the frontline at tissue sites such as the skin lungs and drive the first wave of effector cells generated during the first encounter with pathogen[63–66]. It has been shown that these cells are vital to the initial response against *Klebsiella pneumoniae* infection and derive from exTh17 cells which have lost the expression of their signature cytokine IL-17A[63]. The specific upregulation of CD69 by tissue memory CD4$^+$ and CD8$^+$ T cells in humans and mice suggests that memory T cells differentiating at tissue sites recognize distinct signals to those circulating in blood. While CD8$^+$ T$_{RM}$ within mucosal sites is also characterized by upregulation of the αE integrin, CD103, the role of CD103 on CD4$^+$ T$_{RM}$ is less clear as many CD4$^+$ T$_{RM}$ in mice and humans do not express CD103[33,66–68]. In agreement with these studies, we observed a low proportion of CD4$^+$ T$_{RM}$ cells expressing CD103 (<10%) in the lungs. The increased compartmentalization of CD103$^-$CD69$^+$ CD4$^+$ T$_{RM}$ cells in the lung of mice vaccinated with J8-Lipo-DT-PHAD provides evidence that our vaccine design induces optimal T$_{RM}$ responses. We also observed that lung cells from J8-Lipo-DT-PHAD immunized mice

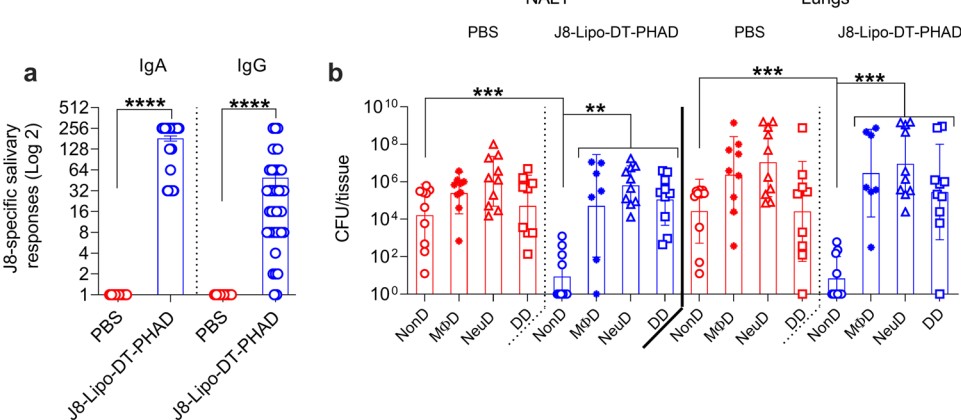

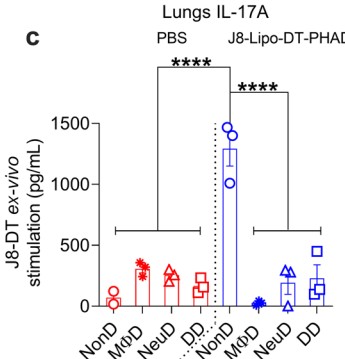

**Fig. 8 | J8-Lipo-DT-PHAD-mediated protection following macrophage and neutrophil depletion.** BALB/c mice (*n* = 10/group; female, 4–6 weeks old) were immunized i.n. with J8-Lipo-DT-PHAD or PBS. One-week post-last-boost J8-specific salivary IgA and IgG (**a**) antibody responses were measured by ELISA and are represented as mean ± SEM. Two weeks after last boost mice were depleted of neutrophils (NeuD), macrophages (MΦD), double depleted (DD), or left non-depleted (NonD). Mice were challenged via the URT with *S. pyogenes* 2031 (5 × 10⁶ cfu/mouse). On day 2 post-challenge all surviving mice were euthanized and NALT and lungs excised to determine *S. pyogenes* burden (**b**), data are represented as the geomean ± geometric SD (cfu/tissue) on a Log 10 scale. The macrophage depletion experiment was repeated once to confirm the results. Statistical analysis was performed using a nonparametric, unpaired Mann-Whitney U test on the raw CFU values (one-tailed) (**p < 0.01; ***p < 0.001; ****p < 0.0001). **c** IL-17A response in the lungs of macrophage and neutrophil-depleted mice. Mice (*n* = 3/group; female, 4–6 weeks old) were immunized i.n. with J8-Lipo-DT-PHAD or PBS. Two weeks after last boost, mice were depleted of neutrophils, macrophages, double depleted, or left non-depleted. Following depletion, mice were euthanized, and lungs excised. Single-cell lung suspensions were stimulated ex-vivo with J8-DT (0.05 mg/well) or media. At 72 h post-stimulation, supernatants were isolated and levels of (**c**) IL-17A were assessed using cytometric bead array as per the manufacturer's instructions. Data are normalized to the media stimulation control and represented as mean ± SEM. Statistical analysis was performed using one-way ANOVA, corrected for multiple comparisons using Dunnett's multiple comparison test to compare against the non-depleted J8-Lipo-DT-PHAD vaccinated cohort (****p < 0.0001). Source data are provided as a Source Data file.

stimulated ex-vivo with J8-DT had significantly more IL-17⁺CD4⁺ (Th17) cells and accordingly significantly higher IL-17A secretion in comparison to PBS and other vaccine configurations that were lacking either J8, DT or PHAD. These findings reveal that vaccination with J8-Lipo-DT-PHAD leads to the induction of J8-DT-specific T_RM and Th17 cells which contribute to the induction of long-term immunity in the URT. Mice vaccinated with J8-Lipo-DT-PHAD were able to induce IL-17A secretion in a J8-DT, J8, and DT-dependent manner which was significantly higher than all other vaccine configurations tested.

Consistent with the role for IL-17A, we observed that removal of CD4⁺ T cells or macrophages and neutrophils (individually or double depleted) from vaccinated mice just prior to infection rendered them unable to control infection. IL-17 has been shown to activate macrophages[69] which are an important source of neutrophil-attracting chemokines, such as CXCL1 and CXCL2, in bacterial infections[56]. CXCL1 produced by tissue-resident macrophages orchestrates neutrophil chemotaxis into the tissue site[57]. Therefore, depletion of macrophages

will abrogate immunity, as neutrophils will not be attracted to the site of infection. Likewise, depletion of neutrophils will also abrogate immunity due to the lack of opsonophagocytic killing of *S. pyogenes*.

Previous data showed that antibodies from mice vaccinated intramuscularly with J8-DT/Alum can transfer protection against skin infection to naive recipients[39,55]. Here, we show that J8-Lipo-DT-PHAD-induced antibodies are functional and can prevent infection (refer to Fig. 4b), but that cellular responses are critical and sufficient.

The primary sites for *S. pyogenes* infection are the URT and skin[70]. Parenteral vaccination alone with J8-DT/Alum is insufficient to provide protection against mucosal streptococcal infection[11]. However, we show here that a mucosal vaccination strategy can induce local protective immunity and that PHAD plays an augmenting role. Utilizing a TLR4 or Myd88 knock-out mouse model to assess the role of PHAD in vaccine-mediated immunity would further support these findings. The adjuvant activity of MPLA/PHAD is attributed primarily to its ability to activate macrophages and induce cytokine cascades[71–73]. Recruited

macrophages can prevent the dissemination of *S. pyogenes* from the infection site into blood and internal organs[20]. Furthermore, an absence of IL-17A has been associated with reduced neutrophil recruitment to the site of infection and *S. pyogenes* clearance significantly attenuated[42]. In this paper, we demonstrate the critical effector function of macrophages and neutrophils in vaccine-induced immunity.

After more than 1-year post-vaccination, i.n. boosting with *S. pyogenes*, led to a significantly increased antibody response in J8-Lipo-DT-PHAD-immunized mice in comparison to mice immunized with J8-Lipo-DT. IL-17A was significantly increased in the lungs and spleen of mice vaccinated with J8-Lipo-DT-PHAD compared to PBS. IL-17A has been shown to play an important role in *S. pyogenes* immunity[48,74]. Recurrent URT infections with *S. pyogenes* have been associated with the upregulation of IL-17A in a TGF-β1 and IL-6-dependent manner[17]. Fan et al., 2014 showed that i.n. vaccination with an *S. pyogenes* cell-wall bound protein, Sortase A, adjuvanted with cholera toxin B subunit (a non-human compliant adjuvant), led to increased CD4+ and IL-17A+ cells in the NALT[75]. Consistent with these reports we found that mice deficient in IL-17A were not protected by vaccination.

Multiple intranasal *S. pyogenes* vaccine candidates have demonstrated an association with antigen-specific salivary IgA and URT protection[11,14,44,76–79]. The primary function of secretory (s) IgA is to stop opportunistic pathogens from infiltrating the mucosal epithelial barriers. This process is referred to as immune exclusion[80]. An early study showed that M6-protein-specific sIgA was able to block adherence to mucosal surfaces and IgG was able to block epitopes responsible for invasion[16]. We have also shown that mucosal IgA specific for a conserved region peptide within the M-protein (p145, which encompasses the sequence of J8) was able to opsonise M5 *S. pyogenes* in-vitro[81]. Here, the necessity of sIgA and mucosal IgG to J8-Lipo-DT-PHAD-mediated immunity has been assessed using pIgR−/− mice. We observed, unexpectedly, that vaccinated pIgR−/− mice had significantly reduced bacterial burden in comparison to PBS mice in tissues following streptococcal challenge, showing that sIgA is not essential to J8-Lipo-DT-PHAD-induced immunity. The importance of sIgA in mucosal immunity for some bacterial and viral mucosal pathogens has been disputed. These include the influenza virus[82], rotavirus[83], *Shigella flexneri*[84], and *Helicobacter pylori*[85]. However, our data do not show that IgA (and IgG) cannot play a role in J8-Lipo-DT-PHAD-induced streptococcal immunity, but rather suggest that any role that mucosal antibody does play would be dependent on the presence of CD4+ T cells, which may function as helper cells for an amnestic antibody response, but also as activators of macrophages. We previously showed that J8-specific IgG can mediate protection against skin and intraperitoneal infections[8,39]. CD4+ T cells played a role in immunity in that model[39]. We observed that depletion of CD4+ T cells just prior to challenge resulted in the mice being unable to maintain J8-specific antibody titers post-challenge resulting in reduced survival[39]. In summary, our data demonstrate that CD4+ T cells are critical for mucosal immunity induced by J8-Lipo-DT-PHAD, consistent with a predominant role in activating macrophages and neutrophils via IL-17 and possibly in also boosting vaccine-specific antibody post *S. pyogenes* exposure. Inclusion of PHAD in the vaccine enables a protective cellular response, alongside long-lived functional antibodies. The vaccine without PHAD (J8-Lipo-DT), whilst protective cannot rely on cellular immune mechanisms involving IL-17A or IL-2, which significantly protect the host from infection. The inclusion of PHAD thus significantly broadens the available protective mechanisms post-vaccination.

We demonstrate that the TLR4 agonist, PHAD, when incorporated into a liposomal intranasal streptococcal vaccine, significantly enhances both cellular and humoral immune responses and the level of protection. The enduring protection seen with the PHAD-containing vaccine is mediated through CD4+ T cells, IL-17A, macrophages, and neutrophils, acting independently of antibody. Vaccine-induced antibodies, which are present at low titers, are functional and likely contribute to the overall level of immunity acting with macrophages and neutrophils. Together, the data highlight a complex interplay of different complementary mechanisms in mucosal immunity to *S. pyogenes* induced by a PHAD-containing liposomal vaccine. This vaccine platform, capable of inducing both cellular and humoral immune responses in the mucosa, could be widely applicable to other mucosal pathogens.

## Methods

### Ethics statement and animals

All animal protocols were reviewed and approved by the Griffith University Animal Ethics Committee (GU-AEC) in accordance with the National Health and Medical Research Council (NHMRC) of Australia guidelines. Experimental protocols involving pIgR knock-out (pIgR−/−; C57BL/6 background)[46], B-cell knock-out (μMT; C57BL/6 background)[47], and IL-17 knock-out (*Il17*−/−; BALB/c background)[86] mice were reviewed and approved by Office of the Gene Technology Regulator (OGTR). BALB/c and C57BL/6 mice (female, 4–6 weeks) were sourced from Animal Resource Centre, Western Australia. *Il17*−/− mice were acquired from Yoichiro Iwakura (Tokyo University of Science, Japan)[86]. Knock-out mice were bred in-house at the Griffith University Animal Facility. All mice were housed in a PC2-certified animal facility in individually ventilated cages (IVC) with a maximum of 5 mice/cage. The relative humidity ranged between 45–65% and temperature at 20–24 °C. Mice were exposed to a 12-h light–dark cycle.

### *S. pyogenes* strains and growth conditions

*S. pyogenes* strain 2031 (*emm* 1, obtained from the Menzies School of Health Research, Darwin, Australia) was serially passaged through mice spleen to enhance its virulence and made streptomycin resistant (200 μg/ml) to enable its identification in throat swabs from mouse commensal bacteria. For all assays, mouse-adapted, streptomycin-resistant 2031 was grown for 16–18 h at 37 °C. Single colonies were inoculated in Todd Hewitt broth (THB; Oxoid, Australia) supplemented with 1% BactoNeopeptone (Thermo Fisher, USA) and yeast. For enumeration of colony forming units (cfu), cultures were plated on Columbia agar containing 5% defibrinated horse blood and 200 μg/ml streptomycin (CBA5%).

### Peptide synthesis

J8-palmitic acid (C16)₂-KSS-J8 was synthesized and purified (>95%) commercially by ChinaPeptides Co. Ltd. (Shanghai, China) and stored lyophilized or in solution at −20 °C[11]. Peptide sequence for (C16)₂-KSS-J8 is: (C16)₂KSSQAEDKVKQSREAKKQVEKALKQLEDKVQ.

### Liposome vaccine formulation

Liposomal vaccines were prepared as per established methods[11,13]. Predetermined amounts of J8-palmitic acid, 3D(6-acyl) PHAD (PHAD; Avanti Polar Lipids Inc., USA) and phospholipids—dipalmitoyl-sn-glycero-3-phosphocholine (DPPC), cholesterol (CHOL) and 1 L-α-phosphatidylglycerol (PG) (Avanti Polar Lipids Inc.) were coated on to round bottom flasks to form a lipid thin film. The thin films were hydrated in 20 mM phosphate-buffered saline (PBS; pH 6) containing trehalose and Diphtheria Toxoid (DT). The resultant liposomal suspension was freeze-dried and vaccines were stored at 4 °C until use. The average particle size (nm) of a liposome was measured using a Nanosizer (Zetasizer Nano Series ZS, Malvern Instruments) as per established method[11]. Homogenous size distribution was confirmed by a low polydispersity index (PDI) of 0.08 for J8-Lipo-DT-PHAD with an average size of 1987 nm (SD 104 nm) before freeze-drying and PDI of 0.16 with an average size of 5213 nm (SD 572.2) after reconstitution of freeze-dried liposomes in PBS. For administration, each vaccine vial was resuspended in PBS, and mice were immunized intranasally (i.n.) with a total volume of 20 μl (30 μg/mouse of J8-palmitic acid, 25 μg/mouse of PHAD and 30 μg/mouse of DT−10 μl/nare).

## Immunization and sample collection

Mice were immunized i.n. as per established method[11]. Mice received a total volume of 20 µl (10 µl/nare) of vaccine or PBS on days 0, 21, and 42. Blood and saliva were collected on day 50 post-primary immunization, as per established protocol[11].

## Enzyme-linked immunosorbent assay (ELISA)

Standard ELISA was used to measure J8-specific serum and salivary antibody levels, as described elsewhere[55]. Briefly, peptide J8 was coated onto NUNC MaxiSorp plates (Thermo Fisher). Serum and saliva samples were diluted 2-fold and antibody levels detected with HRP-conjugated goat anti-mouse IgA antibody (1:1000 dilution; Invitrogen, USA) and HRP-conjugated goat anti-mouse IgG (1:3000 dilution; Bio-Rad, USA). SIGMAFAST OPD substrate was added according to manufacturer's instructions and absorbance was measured at 450 nm on a Tecan Infinite M200 Pro plate reader (Tecan Group Ltd., Switzerland). The end-point titers were defined as the highest dilution that gave an absorbance of >3 standard deviations above the mean absorbance of the negative control wells (containing pooled-sera/saliva, at the same dilution, from PBS-immunized mice).

## Upper respiratory tract (URT) challenge of mice with *S. pyogenes*

Mice were challenged via the URT with $5 \times 10^6$ cfu/mouse of 2031 as per established protocol[11]. Briefly, *S. pyogenes* 2031 (*emm* 1) was administered i.n. at 5 µl/nare (10 µl total) to anaesthetized mice. Mice were monitored daily for signs of illness as per score sheet approved by the GU-AEC.

Throat swabs (TS) and nasal shedding (NS) were collected on days 1–3 post-challenge. For TS, the throats were swabbed using a flocked swab (Copan Diagnostics, USA). Swabs were then suspended in PBS, serially diluted, and dot-plated in duplicate on CBA5% plates. For NS, the nares of mice were pressed onto a CBA5% plates 10 times total (five times/half plate) and exhaled particles were streaked out[87]. On days 2-3 post-infection, mice were sacrificed via $CO_2$ asphyxiation and nasal-associated lymphoid tissue (NALT; functional homolog to human tonsils), and lungs were removed. Tissues were mechanically homogenized in PBS using the Bullet Blender Homogenizer (Next Advance, USA), serially diluted, and dot-plated in duplicate on CBA5% plates. For bacterial enumeration, all plates were incubated at 37 °C overnight, and individual hemolytic streptococcal colonies counted. Mice were monitored twice daily until the experiment end-point in accordance with established clinical scoring system[45]. The mice were scored with the observer blinded to the treatment.

## Preincubation challenge assay

*S. pyogenes* were incubated with neat serum from PBS, J8-Lipo-DT-PHAD, or PBS-immunized BALB/c mice. Following incubation, the bacterial cells were collected by centrifugation and a 10 µL inoculum containing $5 \times 10^6$ cfu/mouse delivered i.n. to naive BALB/c mice. The mice were culled post 72 h of challenge and NALT and lungs harvested and plated onto blood agar plates to determine bacterial burden. The protocol was a modification of a previous assay[8].

## Quantification of secreted cytokines by cytometric bead array

Lung and spleen cells from vaccinated and control mice were prepared in RPMI 1640 medium supplemented with 10% heat-inactivated fetal bovine serum (Invitrogen) and penicillin (100 U ml⁻¹), streptomycin (100 µg ml⁻¹) (Gibco Life Technologies, USA). Cells were stimulated with: J8-DT, DT, J8 at 0.5 mg/mL, heat-killed *S. pyogenes* at MOI of 10, or media alone. Samples were incubated for 72 h in 5% $CO_2$ at 37 °C.

Mouse Th1/Th2/Th17 cytokine kit (Cat: 560485; Becton Dickson, USA) was used according to manufacturers' instructions[11]. Samples were run on a BD LSRFortessa cytometer and data analyzed with FCAP Array (v 3.0 for windows) software.

## Assessment of tissue-resident memory ($T_{RM}$) cells in the lungs

Lung cells were isolated from vaccinated mice and $T_{RM}$ cells were identified from CD3⁺CD4⁺CD62L⁻CD44⁺ effector memory (EM) cells that were CD69⁺CD103⁻. Samples were run on a BD LSRFortessa cytometer and data analyzed with FlowJo v10.8 software (BD Biosciences).

## Intracellular flow cytometry analysis of lung Th17 cells

Lung cells from vaccinated mice were seeded into sterile 96-well v-bottom plates and stimulated with J8-DT (0.05 mg/mL) or media. Following overnight (16 h) incubation at 37 °C with 5% $CO_2$, Brefeldin A (Sigma-Aldrich) was added to the wells at 10 µg/mL, and cells were incubated for an additional 5 h. After incubation, cells were washed in FACS buffer and stained for 30 min at 4 °C for surface markers (CD8a-PECy7, CD3 PE-Dazzle 594, CD62L-PerCP-Cy5.5, live/dead-fixable Near-IR, CD69-BV421, CD4-BV510, CD102-BV711, CD44-BV785 and CD45-BUV395). For intracellular staining, the cells were washed in FACS buffer, permeabilized using the Cytofix/Cytoperm Kit (BD Biosciences), and stained for 45 min at 4 °C with IL-17A-APC. Cells were washed and run on a BD LSRFortessa cytometer and data analyzed with FlowJo v10.8 software (BD Biosciences).

## Cell depletion studies

Macrophages were depleted using Clodronate Liposomes (ClLip; ClodLipBV, Haarlem, The Netherlands). ClLip was delivered intravenously (i.v.; 100 µl at 0.5 mg/mouse) and i.n. (50 µl at 0.25 mg/mouse) for the depletion of systemic and alveolar macrophages, respectively. Control mice received PBS/liposomes. Mice were treated on days −3, −2, −1, and +1, relative to infection on day 0. Depletion was assessed by flow cytometry in lungs (alveolar macrophages; CD45⁺F4/80⁺CD11c⁺) and spleen (MHCII⁺ macrophages; CD45⁺CD11b⁺CD24⁻F4/80⁺MHCII⁺). Depletion was also assessed in paraffin-embedded, NALT, lung, liver, and spleen biopsy samples using F4/80 immunohistochemistry (IHC) staining (QIMR Berghofer histology facility). For IHC staining, ClLip was delivered i.p.+i.n. on days −3, −2 and −1, relative to infection on day 0. Sections were scanned and read at high magnification using ImageScope software (Leica Biosystems, Australia). F4/80⁺ cells were counted in five areas of scanned slides and expressed at the average number of positive cells/high-powered field using ImageJ (National Institutes of Health, Bethesda, USA).

Neutrophils were depleted using anti-Ly6G mAb (clone 1A8, BioXcell). Mice received 100 µL (0.5 mg/mouse) i.p. on days −3. −2 and −1, relative to infection on day 0. Depletion was confirmed by flow cytometry in lungs and spleen (CD45⁺Ly6G⁺CD11b+).

CD4⁺ T cells were depleted using rat anti-mouse CD4 mAb (GK 1.5; Assay Matrix). Mice received 500 µl (0.215 mg/mouse) of anti-mouse CD4⁺ mAb i.p. or Rat IgG (isotype control) on days −4 and −1 before infection (day 0). CD4⁺ depletion was confirmed in cervical lymph nodes, NALT, lungs, and spleen on days 0 and 3 post-mAb administration. A list of antibodies used in all flow cytometry experiments is available in the supplementary materials in Table 1.

Depletion was assessed by flow cytometric analysis using a BD LSRFortessa cytometer and data was analyzed with FlowJo v10.8 software (BD Biosciences).

## Statistics

Statistical analysis was performed with GraphPad Prism 9 software using a nonparametric, unpaired Mann−Whitney $U$ test (one-tailed; 90% confidence interval) to compare test groups, unpaired $t$ test with Welch's correction to compare test groups, one-way ANOVA, corrected for multiple comparisons using Dunnett's multiple comparison test and two-way ANOVA, corrected for multiple comparisons using Sidak's multiple comparison test (ns $p > 0.05$; *$p < 0.05$; **$p < 0.01$; ***$p < 0.001$; ****$p < 0.0001$). ARRIVE guidelines[88] were used to calculate sample size for in-vivo experiments. A sample size of 10 was shown

to provide a power of 0.8 (G*Power) and therefore used for all bacterial challenge animal experiments.

## Reporting summary

Further information on research design is available in the Nature Portfolio Reporting Summary linked to this article.

## Data availability

All data necessary to interpret the findings and draw conclusions about the study are included within the manuscript and supplementary information or from the corresponding author on request. Source data are provided with the manuscript. Source data are provided with this paper.

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

## Acknowledgements

This work was supported by the National Health and Medical Research Council (NHMRC), Australia, Program Grant APP1083548 (M.F.G.), NHMRC Project Grant APP1160379 (M.P.), NHMRC Australia, Investigator Fellowship APP1174091 (M.F.G.), NHMRC Early Career Fellowship APP1069915 (M.Z.), Australian Postgraduate Award (V.O.) and Griffith University Postdoctoral Fellowship (V.O.).

## Author contributions

Conceptualization: V.O., M.Z., A.L., M.G., and M.P. Methodology: V.O., M.Z., A.L., M.G., and M.P. Investigation: V.O., M.Z., A.L., S.E., J.K., J.M., A.C., E.L.L., J.D., and Y.H. Funding acquisition: M.G. and M.P. Project administration: M.G., M.R.B., and M.P. Supervision: M.G. and M.P. Writing—original draft: V.O., M.G., and M.P. Writing—review & editing: V.O., M.G., M.P., A.L., J.K., S.E. and M.Z. Contribution of resources (provision of study materials/animals): G.C.U.

## Competing interests

The authors declare no competing interests.
