## [Peer Review File · Nature Communications]

A Glycolipidated-liposomal peptide vaccine confers long-term mucosal protection against *Streptococcus pyogenes* via IL-17, macrophages and neutrophils.Editorial Note: Parts of this Peer Review File have been redacted as indicated to maintain the confidentiality of unpublished data.

REVIEWER COMMENTS

Reviewer #1 (Remarks to the Author):

There is a significant need for effective vaccines against Streptococcal infections to prevent the acute and chronic sequelae. Systemic vaccines developed by this and other group have preventive efficacy against systemic infections in animal models, but not against mucosal infection. This paper addresses the problem by incorporating a TLR-4 agonist into liposomes along with a lipidated (Palmitic acid x2) form of a B cell epitope (J8) of *S. pyogenes* with Diphtheria toxoid for T cell help and comparing this with liposomes containing J8-lipid-DT as intranasal mucosal and systemic vaccines.

The innovative findings are that:

1. Although i/n delivery of both vaccines had similar protection against *S. pyogenes* in the local NALT and lungs acute 2 week model, the vaccine with the TLR4 agonist PHAD produced more sustained antibody responses at ~450 days, a stronger recall response to *S. pyogenes* and repeat vaccination, and was more protective in the lungs, but not in the NALT, when challenged at 210 days (Fig 4K).
2. Surprisingly the protection with PHAD-J8-lipo-DT vaccine was independent of mucosal IgA and systemic antibodies in the acute 2 week model of infection (fig 5), although serum antibodies did contribute to bacterial killing in vivo (Fig 4M).
3. The protection was dependent on CD4 T cells that were depleted after the development of immune responses to PHAD-J8-lipo-DT vaccine and IL-17 (fig 6), and this was in part dependent on macrophages.

The data are robust, and statistical analysis in general appropriate.

Issues

1. The authors ignore the potential role of neutrophils in either IgG- or IL-17-mediated immunity and attribute the effector mechanisms to be dependent on macrophages. In fact, there is wide variation in the NALT and lung bacterial loads after macrophage depletion (Fig

7L) with overlap between groups, so although this reached significance, it suggests other mechanisms may be in play, such as IgG-enhanced neutrophil responses. Was this experiment repeated? Neutrophil depletion may have a similar effect on vaccine efficacy in this acute 2 week model, and this should be compared with macrophage depletion?

2. The authors claim that other TLR-4 agonists have not been used for mucosal vaccines, but that is not the case and should be corrected eg GLA-SE (PMID 26541135). In addition, vaccines containing the TLR-2/6 agonist, Pam2Cys, conjugated with peptides or proteins are effective as mucosal vaccines against a range of pathogens. Have the authors tested whether the J8-Pam2Cys alone activates TLR-2? Are they combining both TLR4 and TLR-2 activation; this should be discussed.

3. The J8 peptide was combined with DT to provide T cell help, presumably it does not contain T cell epitopes, and T cell responses in vivo are largely to the DT not the J8. But when the authors tested T cell responses ex vivo, "Splenocytes were stimulated ex-vivo with J8-DT (50 µg/well) for 72 h...". This would have re-stimulated T cell responses to both DT and J8. How do they know these are J-8 specific; was DT alone included as a control? Why was the J8 peptide alone not used for re-stimulation? The T cell responses were measured in the spleen, not at the mucosal site in the NALT or lung: of these, T cells in the lung are readily accessible and would have been revealing. The propensity to IL-17 production may be related to the route of immunization rather the addition of PHAD adjuvant.

4. Fig 1 merely shows the wide range bacterial growth at different mucosal sites in different experiments with same challenge dose. This could be in supplementary data. The protective effect of two vaccines are shown in multiple protection experiments in the acute model, but the numbers of replicates for subsequent immunology and challenge experiments are not included: this should be stated in all Figures.

5. Macrophage depletion was quantitated by immunohistochemistry and this declined over between 3 and 5 days. Ideally this depletion should be confirmed by flow cytometry as macrophage populations are not identical at the sites. In addition, the macrophage response at the site of infection within the NALT and lung should be documented with and without vaccine.

Reviewer #2 (Remarks to the Author):

The authors present an investigation in the immune response of mice following vaccination with a liposomal preparation of the Streptococcus pyogenes vaccine candidate J8, with and without the glycolipid PHAD. PHAD is a TLR4 agonist and has been used before (MPLA) in intranasal vaccination; it has also been previously combined in a liposome although not in streptococcal vaccination. In this study, intranasal administration of either vaccine preparation elicited J8 specific mucosal and serum antibodies to varying degrees and were able to reduce bacterial burden following intranasal infection with M1 S. pyogenes in comparison to sham vaccinated mice. Levels of mucosal antibodies were higher in mice vaccinated with the PHAD containing vaccine. Humoral immunity waned beyond a year post infection but boosting with J8 or whole M1 S. pyogenes caused strong antibody responses, indicating good memory responses, which were higher in PHAD vaccinated mice. The finding of prolonged immunity is a strength of this paper. The use of various knockout mouse strains and the depletion of immune cells suggest that the protective capacity of liposomes+PHAD+J8 may not require antibody responses, though the lack of controls made this hard to interpret. The authors conclude that PHAD significantly enhances cellular and humoral immune responses and protection against challenge, in a CD4+ T cell, macrophage and IL-17 dependent manner however there are key issues that need addressing

1.Except for data in figure 2 and the long-term booster experiment, all the comparisons seem to be between the new adjuvanted liposomes and saline rather than new adjuvanted liposomes and liposomes. So apart from the experiment in Figure 2, none of the other comparisons provide a valid comparison to address the key question of whether the novel component of the vaccine is responsible for superior protection.

2.Considering the high bacterial burden found in the lungs of these animals and the high clinical scores reported in the study, this model appears to be invasive/lower respiratory tract rather than upper respiratory tract challenge- the authors should acknowledge this.

3. The comparisons undertaken are unorthodox- the authors quite correctly present CFU from individual mice, but then compare the groups based on an entirely different set of metrics- they assign a % reduction in bacterial load- comparing individual mice with a single value obtained from the control group (geometric mean)- this negates the any variance in the control group, yet this variation is important to take into account- indeed many of the CFU counts do not seem that different. Confusingly the p values for the '% reduction' comparison is then overlaid on the graphs showing CFU, creating the impression that the CFU counts themselves are significantly different, although they may not be?

4. The authors examine mechanism of vaccine-induced protection in several ways (mice lacking antibodies, mice lacking IL17, CD4 depletion, etc). However, in each case, they have not undertaken the correct control group comparison. It is entirely possible that adjuvanted liposomes will confer greater protection than PBS alone - even in mice that lack antibodies. But it isn't clear that we are seeing a 'vaccine' effect here, or some other 'boosted' non specific aspect of innate immunity. Surely the control comparison would have been with liposomes+DT?

5. I could not see an experiment comparing the antibody negative mice with wild type mice- this would allow us to see what contribution antibodies might make?

6. It would have been clearer to further examine the mechanism of protection in antibody-negative mice by depleting IL17 or CD4 cells in those same mice (or crossing with appropriate KO mice rather than exploring their role in separate mouse strains. As it stands, it would appear that IL-17 and CD4 cells contribute to boosted immunity to *S. pyogenes*, but we are not sure what the relative contribution of these components is?

Minor comments

Figure 1

oAs this figure is establishing the model and providing justification for only including NALT and lung bacterial burdens throughout the rest of the paper the authors should consider

moving it to the supplementary materials, or considering the data is made up of control groups represented throughout the rest of the investigation this figure could be omitted altogether

oWere throat swabs and nasal shedding measured for all mouse infections shown throughout the paper? If so, these data should be included either in the substantive figures or in the supplementary materials

Figure 2

oA and B - These two panels could benefit from being presented on their own as the first figure (minor issue, but the use of J8SSK and KSSJ8 is a bit confusing at first read)

oDescription of statistics performed is incorrect given that there are comparisons between more than just the PBS group and test groups presented in panels C-E

oThe authors should consider presenting bacterial burden as box plots to better show the distribution of data within groups

oWas the experiment in figure 2 adequately powered- if so then a single experiment can provide the answer required

Figure 3

oThis figure is purely descriptive, as well as confusing and repeats data shown throughout the rest of the study. If the authors believe that the figure is necessary, they should consider including it in the supplementary materials and presenting the summary data as below (NB, no statistical difference between J8-Lipo-DT-PHAD vs J8-Lipo-DT-PHAD, so while PHAD may be enhancing specific immune responses, it is not enhancing protection against challenge)

Figure 4

oThere is a lot of information presented in this figure and the authors may want to consider revising in order to help with readability.

oThe timeline is unclear as A shows day 0 as first dose of vaccine, but then presumably switches to days post last dose of vaccine at some point, as day 455 is shown in B-I which specifies time post last vaccine boost. The timeline also seems to imply that all mice were subjected to URT challenge on day 252?

oDescription of stats implies only comparisons to PBS control group however there are clearly comparisons within and between test groups. As the study purports that PHAD enhances immunity, comparisons between test groups are necessary, and panels Bi, Ci, Di, Ei do support that claim.

oF-G only shows the data for mice vaccinated with J8-Lipo-DT-PHAD, it would be useful to be able compare these data against those of the J8-Lipo-DT as the purpose of this figure appears to be that PHAD enhances immunity compared to the liposomes not containing PHAD

olt is unclear why two different boosts (*S. pyogenes* vs J8-DT) were used

Figure 5

oWhile the plgR deficient mice lack mucosal antibodies, based on the CFU seen in the lungs of the control group is it not possible that the protection seen is in fact due to the serum antibodies present protecting against pneumonia? In which case the μ MT mice provide a more compelling argument that it is the cellular response following vaccination with J8-Lipo-DT-PHAD that is critical for protection?

oCan the authors comment on whether these same results be seen when vaccinating with J8-Lipo-DT? Could the effects be a non-specific effect of the liposomes+DT+PHAD?

ol could not see an experiment comparing the antibody negative mice with wild type mice- this would allow us to see what contribution antibodies might make?

Figure 7

oPanels validating depletion can be moved to supplementary materials and J-L could be combined with Figure 6

Reviewer #3 (Remarks to the Author):

In this manuscript submitted by Ozberk, et al., the authors hypothesize that a TLR agonist 3D(6-acyl) PHAD will improve the ability of a vaccine directed against a 12-mer derived from the *S. pyogenes* M protein (J8) to induce mucosal protection. To this end they evaluated a delivery system consisting of liposomes encapsulating DT and displaying lipidated J8 that did

or did not contain PHAD. For the purposes of this study the authors demonstrated that following upper respiratory challenge with *S. pyogenes* 2031, colonization levels were considerably higher in the isolated NALT and lungs than in particles exhaled from the nose or on throat swabs. They demonstrate that salivary IgA and IgG levels as well as serum IgA (but not IgG) are higher following immunization with vaccine containing PHAD than in vaccine that did not contain PHAD, and importantly that vaccine containing PHAD offered significantly better protection in the NALT, although a difference in the lungs was not significant, following URT challenge. Immunity following immunization with the PHAD containing vaccine was long-lived with higher salivary and serum Ig levels observed over 428 days with stronger recall response to a booster dose, although at this late time point, protection measured by clinical score and bacterial burden in the NALT and lungs was not significantly different between animals that receive vaccine with and without PHAD. The authors then seek to evaluate the mechanism of vaccine mediated protection in the respiratory tract and find that protection as well as J8-DT mediated cytokine production in the spleen in vaccinated mice is maintained in both pIgR and uMT mice suggesting that antibody is not required for protection. To determine whether vaccine mediated protection requires CD4+ T cells and/or IL-17 they ask whether antibody-mediated depletion of CD4 cells or immunization of mice lacking IL-17 interferes with protection of the respiratory tract. Despite maintenance of Ig responses in both case, protection was strongly abrogated, suggesting a role for Th17 cells in vaccine mediated protection of the respiratory tract. Finally, the authors suggest that this IL-17 mediated protection requires macrophages, as clodronate liposome-mediated depletion of previously immunized mice interferes with protection.

The experiments described in this study appear well done and the manuscript is written clearly. Understanding immune responses that protect the respiratory tract from bacterial pathogens including *S. pyogenes* following vaccination is of high relevance to human health and is incompletely understood. That said there are a number of significant issues that limit the potential impact of this study and I have described these under major issues. I have also included a list of minor issues that while not significantly affecting the impact of the paper could potentially improve the quality of the data presented, if addressed.

Major Issues:

1. The observation that protection of the respiratory tract from colonization with *S. pyogenes* following immunization depends on IL-17 is interesting, but a more detailed characterization of underlying protective mechanisms would increase impact. There are now numerous examples of extracellular pathogens (pneumococcus, Staph, Klebsiella, candida; reviewed in Iwanaga N, Kolls JK. *Immunology*. 2019 Jan;156(1):3-8.) whose elimination requires IL-17. Thus this observation in and of itself may have limited additional impact given that there are already several other publications involving *S. pyogenes* suggesting a similar phenomenon (referenced by the authors). However, understanding the mechanisms of IL-17 mediated protection of the respiratory tract is quite important and while the authors do provide some data regarding macrophages (see comments below) a key question is whether protection is mediated by local resident memory T cells within the mucosa, and if the PHAD containing vaccine more efficiently induces localization of T cells to NALT or the respiratory mucosa than vaccines lacking PHAD. One of the reported advantages of mucosal rather than parenteral immunization is the ability of mucosally administered vaccines to induce resident memory, and therefore understanding this issue in additional detail would contribute important mechanistic insights to the manuscript.

2. The authors should more clearly define the role of DT in this system and include experiments with vaccine lacking DT. The authors suggest that DT is providing "T cell help", which may be important for antibody production. However, given that the important end-point here appears to be a cell mediated response to the J8-peptide (rather than DT) the relevant helper response appears to be to J8 itself, leading to the question of whether DT is acting as a critical adjuvant independent to a role as a carrier protein providing help, or is in fact potentially not necessary.

3. The authors clearly demonstrate the PHAD enhances vaccine mediated protection, and the authors hypothesize that this is based on TLR4-mediated adjuvant effects, but this mechanism requires further evaluation. Certainly, demonstrating loss of efficacy in a TLR4 or Myd88 knockout mouse would add important mechanistic detail, as would further insights into whether TLR4 agonists enhance the generation of protective mucosal versus

(potentially non-protective) systemic Th17 responses. Additional details regarding the cell-specific requirement for TLR4 signaling on different sets of APCs for effective mucosal vaccination could be novel and increase impact.

4. Clodronate-mediated macrophage depletion experiments are interesting but further evaluation of the underlying mechanisms would add to the manuscript. A key question is whether depletion of macrophages interferes with the IL-17 response itself or rather leave IL-17 responses intact, suggesting as the authors propose that macrophages are potential effectors of IL-17 mediated responses in this system. Identifying increased expression of IL-17 regulated genes such as Cxcl1 in macrophages isolated from vaccinated mice following challenge would further support a critical role for macrophages as effectors of the Th17 response.

Minor Issues:

1. It is not clear what each dot represents in Figure 1. Does each dot represent the mean (or geomean) of one of 6 individual experiments? How many mice does each dot represent and how many mice in total are being evaluated?

2. The data in 2E is expressed as geomean +/- SD, but then significance is tested using Mann-Whitney. If data is non-parametric, median and interquartile range are more appropriate descriptive statistics

3. Figure 3 summarizes a large amount of data much of which is not clearly described in the text. For instance, there are several different mouse strains included without a clear rationale. Further, similar to the data in Figure 2, if using Mann-Whitney, mean and SD are not appropriate descriptive statistics. Also, exactly what "percent reduction" in these comparisons refers to and whether it is a statistically valid and/or requires confidence intervals is not well defined.

4. It would be helpful to describe methodology of boosting used for the data displayed in Figure 4.

5. In Figure 5 the authors demonstrate that protection from colonization following vaccination remains intact in mice lacking mucosal or total antibodies. This data would be enhanced by demonstrating the maintenance of both systemic and mucosal IL-17 responses to Ag rechallenge in these mice.

6. Data in figure 6 would be stronger if the protective response to vaccination was directly compared in WT and IL-17-deficient mice.

Reviewer#1 (Remarks to the author)

There is a significant need for effective vaccines against Streptococcal infections to prevent the acute and chronic sequelae. Systemic vaccines developed by this and other group have preventive efficacy against systemic infections in animal models, but not against mucosal infection. This paper addresses the problem by incorporating a TLR-4 agonist into liposomes along with a lipidated (Palmitic acid x2) form of a B cell epitope (J8) of *S. pyogenes* with Diphtheria toxoid for T cell help and comparing this with liposomes containing J8-lipid-DT as intranasal mucosal and systemic vaccines.

The innovative findings are that:

1. Although i/n delivery of both vaccines had similar protection against *S. pyogenes* in the local NALT and lungs acute 2 week model, the vaccine with the TLR4 agonist PHAD produced more sustained antibody responses at ~450 days, a stronger recall response to *S. pyogenes* and repeat vaccination, and was more protective in the lungs, but not in the NALT, when challenged at 210 days (Fig 4K).
2. Surprisingly the protection with PHAD-J8-lipo-DT vaccine was independent of mucosal IgA and systemic antibodies in the acute 2 week model of infection (fig 5), although serum antibodies did contribute to bacterial killing in vivo (Fig 4M).
3. The protection was dependent on CD4 T cells that were depleted after the development of immune responses to PHAD-J8-lipo-DT vaccine and IL-17 (fig 6), and this was in part dependent on macrophages.

The data are robust, and statistical analysis in general appropriate.

Reviewer #1 (major comments):

1. The authors ignore the potential role of neutrophils in either IgG- or IL-17-mediated immunity and attribute the effector mechanisms to be dependent on macrophages. In fact, there is wide variation in the NALT and lung bacterial loads after macrophage depletion (Fig 7L) with overlap between groups, so although this reached significance, it suggests other mechanisms may be in play, such as IgG-enhanced neutrophil responses. Was this experiment repeated? Neutrophil depletion may have a similar effect on vaccine efficacy in this acute 2-week model, and this should be compared with macrophage depletion?

Thank you for your comment. To answer this question, we have repeated the macrophage depletion experiment and compared it to neutrophil depletion. Consistent with our previous data, we found that macrophage depletion impaired protection mediated by J8-Lipo-DT-PHAD in the NALT and lungs (Figure 7H). We also showed that neutrophil depletion and double depletion (macrophages and neutrophils) impaired J8-Lipo-DT-PHAD-mediated protection (Figure 7H). Vaccinated non-depleted mice induced significantly more IL-17A in the lungs following *ex-vivo* stimulation with J8-DT (Figure 7I). This effect was abrogated either when macrophage, neutrophil or both subsets were depleted, suggesting that both macrophages and neutrophils play an important and complimentary role in vaccine-mediated

protection via modulation of IL-17A levels. While neutrophils are known to produce IL-17A, macrophages can indirectly stimulate IL-17A production via IL-23.

2. *The authors claim that other TLR-4 agonists have not been used for mucosal vaccines, but that is not the case and should be corrected eg GLA-SE (PMID 26541135). In addition, vaccines containing the TLR-2/6 agonist, Pam2Cys, conjugated with peptides or proteins are effective as mucosal vaccines against a range of pathogens. Have the authors tested whether the J8-Pam2Cys alone activates TLR-2? Are they combining both TLR4 and TLR-2 activation; this should be discussed.*

We apologize for claiming that other TLR4 agonists have not been used for mucosal vaccines. We have now corrected our statement. Please refer to manuscript, page 4, lines 101-104.

In Zaman, Ozberk et. al. 2016 (PMID: 27976706) we showed that i.n. vaccination with J8-Pam2Cys does not elicit an immune response and that the Lipo-DT-PHAD vaccine platform is required for the induction of an antibody response (refer to Figure 2A in the published manuscript). We have not specifically tested whether there is a combination of a TLR2+TLR4 response; lipopeptides do engage TLR2 and this, together with liposomal activation of the NLRP3 inflammasome may be critical to innate immune stimulation.

3. *The J8 peptide was combined with DT to provide T cell help, presumably it does not contain T cell epitopes, and T cell responses in vivo are largely to the DT not the J8. But when the authors tested T cell responses ex vivo, "Splenocytes were stimulated ex-vivo with J8-DT (50 µg/well) for 72 h...". This would have re-stimulated T cell responses to both DT and J8. How do they know these are J8 specific; was DT alone included as a control? Why was the J8 peptide alone not used for re-stimulation? The T cell responses were measured in the spleen, not at the mucosal site in the NALT or lung: of these, T cells in the lung are readily accessible and would have been revealing. The propensity to IL-17 production may be related to the route of immunization rather the addition of PHAD adjuvant.*

The T-cell response in splenocytes in response to *ex-vivo* stimulation with J8-DT, DT and J8 (peptide alone) were measured and data for IL-17A are represented in Figure 3F, for IL-6 in Supp. figure 3A and for IL-2 in Supp. figure 3B. We show that J8-DT stimulation induces significantly more IL-17A, IL-6 and IL-2 in the immunised mice at day 50 (I-Day 50; 1-week post-last vaccine boost) when compared to DT and J8 alone. J8 alone induces IL-6 and IL-2 levels in mice that were immunised-rested for over one year and boosted i.n. *in-vivo* with J8-DT (I-R Boosted J8-DT; Supp. figure 3B-C). This cytokine response is comparable to the response with J8-DT. DT stimulation led to an increased response in IL-6 in mice that were immunised-rested for one year and boosted i.n. with *S. pyogenes* (I-R Boosted StrepA; Supp. figure 3B). It is possible that there is some cross-reactivity between StrepA and DT. Together, these data propose a complementary and non-redundant role of J8 and DT in underpinning the Th1/2/17 responses induced by J8-Lipo-DT-PHAD vaccination.

We have also measured the IL-17A response in the lungs of vaccinated mice (Figure 1G). Mice immunised with J8-Lipo-DT-PHAD show significantly more IL-17A in comparison to all other vaccine cohorts (Figure 1G). We believe the IL-17A response is due to both the 'complete' vaccine (J8-Lipo-DT-PHAD) and the i.n. route of immunization. The i.n. route of vaccination

leading to a local IL-17A response has been demonstrated in the publication by Christensen *et. al.* 2017 (PMID: 27049058).

4. Fig 1 merely shows the wide range bacterial growth at different mucosal sites in different experiments with same challenge dose. This could be in supplementary data. The protective effect of two vaccines are shown in multiple protection experiments in the acute model, but the numbers of replicates for subsequent immunology and challenge experiments are not included: this should be stated in all Figures.

Figure 1 has been moved to Supp. figure 1.

The numbers of replicates for all experiments have been incorporated into the figure legends.

5. Macrophage depletion was quantitated by immunohistochemistry and this declined over between 3 and 5 days. Ideally this depletion should be confirmed by flow cytometry as macrophage populations are not identical at the sites. In addition, the macrophage response at the site of infection within the NALT and lung should be documented with and without vaccine.

Thank you for your suggestion. We agree that identifying the macrophage populations at the site of infection will add significant information to the current work. We have now characterised macrophage populations in the lungs and spleen of vaccinated and control (PBS immunised) mice by flow cytometry. We observed that treatment with Clodronate Liposomes depleted mainly alveolar macrophages (AM) in the lungs (defined as CD45⁺F4/80⁺CD11c⁺ cells) of both vaccinated and control mice (Figure 7Aii and D). Following depletion, there was approximately 50% of AM reduction in the control group (PBS-treated) and 40% of reduction in the vaccinated group in comparison to the non-depleted cohorts (Figure 7D). AM have been reported as the first sentinels of the respiratory tract and play a very important role against respiratory tract infections (reviewed by Allard *et. al.* PMID: 30108592). In parallel, we have now observed around 85% reduction of mature macrophages in the spleen (defined as CD45⁺F4/80⁺MHCII⁺ cells) following clodronate treatment in both vaccinated and control groups (Figure 7F).

Reviewer#2 (Remarks to the author)

The authors present an investigation in the immune response of mice following vaccination with a liposomal preparation of the *Streptococcus pyogenes* vaccine candidate J8, with and without the glycolipid PHAD. PHAD is a TLR4 agonist and has been used before (MPLA) in intranasal vaccination; it has also been previously combined in a liposome although not in streptococcal vaccination. In this study, intranasal administration of either vaccine preparation elicited J8 specific mucosal and serum antibodies to varying degrees and were able to reduce bacterial burden following intranasal infection with M1 *S. pyogenes* in comparison to sham vaccinated mice. Levels of mucosal antibodies were higher in mice vaccinated with the PHAD containing vaccine. Humoral immunity waned beyond a year post infection but boosting with J8 or whole M1 *S. pyogenes* caused strong antibody responses, indicating good memory responses, which were higher in PHAD vaccinated mice. The finding of prolonged immunity is a strength of this paper. The use of various knockout mouse strains

and the depletion of immune cells suggest that the protective capacity of liposomes+PHAD+J8 may not require antibody responses, though the lack of controls made this hard to interpret. The authors conclude that PHAD significantly enhances cellular and humoral immune responses and protection against challenge, in a CD4+ T cell, macrophage and IL-17 dependent manner however there are key issues that need addressing.

Reviewer #2 (major comments):

1. Except for data in figure 2 and the long-term booster experiment, all the comparisons seem to be between the new adjuvanted liposomes and saline rather than new adjuvanted liposomes and liposomes. So apart from the experiment in Figure 2, none of the other comparisons provide a valid comparison to address the key question of whether the novel component of the vaccine is responsible for superior protection.

To address this, we have repeated immunogenicity and protection studies with all the relevant controls. We have conducted two experiments to include all the relevant control groups. In the first experiment (Figure 1C-D), we have included the vaccine groups J8-Lipo-DT-PHAD and J8-Lipo-DT and the control groups Lipo-DT-PHAD, Lipo-DT (liposomes without J8) and PBS (saline). In the second experiment (Figure 1E-F), we have included the vaccine groups J8-Lipo-DT-PHAD and J8-Lipo-DT and the control groups J8-Lipo-PHAD, J8-Lipo (liposomes without DT/PHAD) and PBS (saline). In both the experiments, we showed that there is no significant difference in protection between the mice vaccinated with saline vs. mice vaccinated with the liposome controls Lipo-DT-PHAD, Lipo-DT, J8-Lipo-PHAD or J8-Lipo. Therefore, for all following studies we used saline only as a negative control.

2. Considering the high bacterial burden found in the lungs of these animals and the high clinical scores reported in the study, this model appears to be invasive/lower respiratory tract rather than upper respiratory tract challenge- the authors should acknowledge this.

This is now acknowledged. Please refer to page 4, lines 129-130 of the manuscript.

3. The comparisons undertaken are unorthodox- the authors quite correctly present CFU from individual mice, but then compare the groups based on an entirely different set of metrics- they assign a % reduction in bacterial load- comparing individual mice with a single value obtained from the control group (geometric mean)- this negates the any variance in the control group, yet this variation is important to take into account- indeed many of the CFU counts do not seem that different. Confusingly the p values for the '% reduction' comparison is then overlaid on the graphs showing CFU, creating the impression that the CFU counts themselves are significantly different, although they may not be?

We agree that the variation within a group is important and must be identified, that is why immunogenicity and protection data are always presented as a bar graph (geomean \pm geometric SD) and as well each individual mouse is represented as a scatter dot-plot on the graph. The percent geomean reduction is used in the manuscript only as a tool to describe the data in addition to the figure. The P values presented on top of the graph always are in reference to the CFU counts themselves. The % reduction is in addition to the P values, and

it is calculated based on the geomean of the PBS (control) group vs. the geomean of the experimental group.

Statistical analysis was performed with GraphPad Prism 9 software using a nonparametric, unpaired Mann-Whitney U test (one-tailed; 90% confidence interval) to compare test groups to the PBS control group and two-way ANOVA, corrected for multiple comparisons using the Sidak's multiple comparison test (ns $p > 0.05$; * $p < 0.05$; ** $p < 0.01$; *** $p < 0.001$; **** $p < 0.0001$). Geomean % reduction is used in addition to the statistical methods outlined above or when statistical significance is not reached.

4. The authors examine mechanism of vaccine-induced protection in several ways (mice lacking antibodies, mice lacking IL17, CD4 depletion, etc). However, in each case, they have not undertaken the correct control group comparison. It is entirely possible that adjuvanted liposomes will confer greater protection than PBS alone - even in mice that lack antibodies. But isn't clear that we are seeing a 'vaccine' effect here, or some other 'boosted' non specific aspect of innate immunity. Surely the control comparison would have been with liposomes+DT?

"In figure 1 of the manuscript, we conducted immunogenicity and protection studies with the following vaccine combinations: J8-Lipo-DT-PHAD, J8-Lipo-DT, Lipo-DT-PHAD, Lipo-DT, J8-Lipo-PHAD, J8-Lipo or PBS (saline). We showed that there is no significant difference in protection between the mice vaccinated with saline vs. mice vaccinated with Lipo-DT-PHAD, Lipo-DT, J8-Lipo-PHAD or J8-Lipo. Therefore, for all subsequent studies we used saline only as a negative control."

5. I could not see an experiment comparing the antibody negative mice with wild type mice- this would allow us to see what contribution antibodies might make?

We thank the reviewer for this important comment. We have now compared vaccinated pIgR^{-/-} mice with vaccinated wild-type (WT) mice and vaccinated MuMT mice with vaccinated WT mice for protection in NALT and lungs and observed no significant difference in the level of protection. However, we are aware that knock-out (KO) mice may have other compensatory mechanisms making it difficult to fully exclude a role for antibodies based on this experimental model. It is worth noting that vaccinated KO are protected compared to the control KO mice in all comparisons (pIgR and MUMT experiments). These comments have now been included in the manuscript (please refer to page 9, lines 278-282 of the manuscript).

6. It would have been clearer to further examine the mechanism of protection in antibody-negative mice by depleting IL17 or CD4 cells in those same mice (or crossing with appropriate KO mice rather than exploring their role in separate mouse strains. As it stands, it would appear that IL-17 and CD4 cells contribute to boosted immunity to S. pyogenes, but we are not sure what the relative contribution of these components is?

Thank you for your suggestion. In the new data included in the manuscript we have shown that Th17 CD4⁺ cells and neutrophils are significantly higher in the NALT of J8-Lipo-DT-PHAD immunized mice in comparison to PBS mice (Supp. Figure 6A-B). We have also shown the dominance of Th17 T_{RM} cells in the lungs of J8-Lipo-DT-PHAD immunized mice in comparison

to PBS mice (Figure 6F). We believe that protection is mediated through CD4+ Th17 T_{RM} cells in the URT tissue through macrophages and neutrophils and that vaccine antibodies likely contribute to overall level of immunity acting with macrophages and neutrophils.

We will take this suggestion on board and include these studies in our future work.

Reviewer #2 (minor comments):

Figure 1

As this figure is establishing the model and providing justification for only including NALT and lung bacterial burdens throughout the rest of the paper the authors should consider moving it to the supplementary materials, or considering the data is made up of control groups represented throughout the rest of the investigation this figure could be omitted altogether.

Figure 1 has been moved to supplementary materials. Please refer to supp. figure 1.

Were throat swabs and nasal shedding measured for all mouse infections shown throughout the paper? If so, these data should be included either in the substantive figures or in the supplementary materials.

Throat swabs (TS) and nasal shedding (NS) were included for all mouse experiments. The data from TS and NS are semi-qualitative readouts and due to the nature of procedural techniques lead inconsistent data. Overall, these data were deemed unreliable and therefore, we have not presented this data in the manuscript.

Figure 2

A and B - These two panels could benefit from being presented on their own as the first figure (minor issue, but the use of J8SSK and KSSJ8 is a bit confusing at first read).

Figure 2 is now figure 1 in the manuscript. We have presented the peptides as J8SSK and KSSJ8 because this is the way the peptides were published in Zaman, Ozberk *et. al.* 2016 (PMID: 27976706). We have kept this the same as we refer to this publication numerous times throughout the manuscript.

Description of statistics performed is incorrect given that there are comparisons between more than just the PBS group and test groups presented in panels C-E.

We have used Mann-Whitney U test to compare between two groups. In all cases, we are comparing the J8-Lipo-DT-PHAD vaccinated group to the other vaccine combinations listed in the figure. The other groups are 'reference' groups. Since it is a group vs. group hypothesis that is being tested, we chose the Mann Whitney U test statistical analysis.

The authors should consider presenting bacterial burden as box plots to better show the distribution of data within groups.

We have presented the immunogenicity and protection data in the manuscript as bar + scatter dot-plot as it clearly defines the geometric mean \pm geometric SD within each group

while also showing each individual mouse data point. We believe that this method is a clear representation of the distribution of data within groups.

Was the experiment in figure 2 adequately powered- if so then a single experiment can provide the answer required.

The experiment was adequately powered. For power calculations a Shapiro-Wilk test was used to calculate that the data was not normally distributed. The parent distribution input parameter for the power analysis was logistic due to the heavy tails on the distribution. Based on data from 5 separate challenge experiments and the assumptions (listed below), an average group number of **10** will provide a medium effect size and a power of 0.8.

For our statistical analysis, our assumptions were as below;

- Power of 0.8.
- Alpha value of 0.05.
- Two sided (To test **both** if the mean is significantly greater than x and if the mean significantly less than x).
- Not normally distributed.

Figure 3

This figure is purely descriptive, as well as confusing and repeats data shown throughout the rest of the study. If the authors believe that the figure is necessary, they should consider including it in the supplementary materials and presenting the summary data as below (NB, no statistical difference between J8-Lipo-DT-PHAD vs J8-Lipo-DT-PHAD, so while PHAD may be enhancing specific immune responses, it is not enhancing protection against challenge).

Figure 3 is now figure 2 in the updated manuscript. This figure is included in the manuscript to provide an overview of immunity across all experiments described in this paper. We believe that this figure is necessary as it shows the distribution of data between experiments and it is an un-bias representation of the protection afforded by J8-Lipo-DT and J8-Lipo-DT-PHAD vaccination in comparison to PBS.

Figure 4

There is a lot of information presented in this figure and the authors may want to consider revising in order to help with readability.

Figure 4 is now figure 3 in the updated manuscript. We have now only included IL-17A cytokine data in this figure panel and removed the IL-6 and IL-2 cytokine data to Supp. Figure 3. This has reduced the number of graphs within the figure panel to improve readability.

The timeline is unclear as A shows day 0 as first dose of vaccine, but then presumably switches to days post last dose of vaccine at some point, as day 455 is shown in B-I which specifies time post last vaccine boost. The timeline also seems to imply that all mice were subjected to URT challenge on day 252?

We apologize for this mislabelling. We have changed the x-axis title on figures B-E to time (days) to match the timeline (figure A). We have also indicated on the timeline the number of mice subjected to URT challenge on day 252.

Description of stats implies only comparisons to PBS control group however there are clearly comparisons within and between test groups. As the study purports that PHAD enhances immunity, comparisons between test groups are necessary, and panels Bi, Ci, Di, Ei do support that claim.

The statistical description has been updated. Please refer to the Figure 3 legend.

F-G only shows the data for mice vaccinated with J8-Lipo-DT-PHAD, it would be useful to be able compare these data against those of the J8-Lipo-DT as the purpose of this figure appears to be that PHAD enhances immunity compared to the liposomes not containing PHAD.

F-G have been moved to supplementary figures (Supp. figure 3). J8-Lipo-DT has not been shown as there was no cellular response detected for this vaccine.

It is unclear why two different boosts (*S. pyogenes* vs J8-DT) were used.

Two different boosts were used to compare the boosting capacity of *S. pyogenes* (to model natural boosting via infection) and J8-DT (to model antigen boosting).

Figure 5

While the plgR deficient mice lack mucosal antibodies, based on the CFU seen in the lungs of the control group is it not possible that the protection seen is in fact due to the serum antibodies present protecting against pneumonia? In which case the μ MT mice provide a more compelling argument that it is the cellular response following vaccination with J8-Lipo-DT-PHAD that is critical for protection?

We agree with the reviewer's interpretation of the data. We first used plgR^{-/-} mice to assess the role of mucosal antibodies in J8-Lipo-DT-PHAD-mediated immunity. When we saw that vaccinated plgR^{-/-} mice were protected we used MuMT mice to assess the role of antibodies (mucosal and systemic) in protection. The MuMT data confirmed that the cellular response following vaccination is a contributor to protection.

Can the authors comment on whether these same results be seen when vaccinating with J8-Lipo-DT? Could the effects be a non-specific effect of the liposomes+DT+PHAD?

We have not assessed J8-Lipo-DT-mediated protection in plgR^{-/-} or MuMT mice. We do not think that liposomes+DT+PHAD provide a non-specific effect. Data from wild-type BALB/c mice show that protection afforded by liposomes+DT+PHAD is comparable to PBS (figure 2D).

I could not see an experiment comparing the antibody negative mice with wild type mice- this would allow us to see what contribution antibodies might make?

Please refer to reviewer#2 (major comments) comment 5.

Figure 7

Panels validating depletion can be moved to supplementary materials and J-L could be combined with Figure 6.

Figure 7 has now been revised to include flow cytometry analysis of depletion. The IHC panels validating depletion have been moved to supplementary figures (Supp. figure 7).

Reviewer#3 (Remarks to the author)

In this manuscript submitted by Ozberk, et al., the authors hypothesize that a TLR agonist 3D(6-acyl) PHAD will improve the ability of a vaccine directed against a 12-mer derived from the *S. pyogenes* M protein (J8) to induce mucosal protection. To this end they evaluated a delivery system consisting of liposomes encapsulating DT and displaying lipidated J8 that did or did not contain PHAD. For the purposes of this study the authors demonstrated that following upper respiratory challenge with *S. pyogenes* 2031, colonization levels were considerably higher in the isolated NALT and lungs than in particles exhaled from the nose or on throat swabs. They demonstrate that salivary IgA and IgG levels as well as serum IgA (but not IgG) are higher following immunization with vaccine containing PHAD than in vaccine that did not contain PHAD, and importantly that vaccine containing PHAD offered significantly better protection in the NALT, although a difference in the lungs was not significant, following URT challenge. Immunity following immunization with the PHAD containing vaccine was long-lived with higher salivary and serum Ig levels observed over 428 days with stronger recall response to a booster dose, although at this late time point, protection measured by clinical score and bacterial burden in the NALT and lungs was not significantly different between animals that receive vaccine with and without PHAD. The authors then seek to evaluate the mechanism of vaccine mediated protection in the respiratory tract and find that protection as well as J8-DT mediated cytokine production in the spleen in vaccinated mice is maintained in both pIgR and uMT mice suggesting that antibody is not required for protection. To determine whether vaccine mediated protection requires CD4+ T cells and/or IL-17 they ask whether antibody-mediated depletion of CD4 cells or immunization of mice lacking IL-17 interferes with protection of the respiratory tract. Despite maintenance of Ig responses in both case, protection was strongly abrogated, suggesting a role for Th17 cells in vaccine mediated protection of the respiratory tract. Finally, the authors suggest that this IL-17 mediated protection requires macrophages, as clodronate liposome-mediated depletion of previously immunized mice interferes with protection.

The experiments described in this study appear well done and the manuscript is written clearly. Understanding immune responses that protect the respiratory tract from bacterial pathogens including *S. pyogenes* following vaccination is of high relevance to human health and is incompletely understood. That said there are a number of significant issues that limit the potential impact of this study and I have described these under major issues. I have also included a list of minor issues that while not significantly affecting the impact of the paper could potentially improve the quality of the data presented, if addressed.

Reviewer #3 (major comments)

1. The observation that protection of the respiratory tract from colonization with *S. pyogenes* following immunization depends on IL-17 is interesting, but a more detailed characterization of underlying protective mechanisms would increase impact. There are now numerous examples of extracellular pathogens (*pneumococcus*, *Staph*, *Klebsiella*, *candida*; reviewed in Iwanaga N, Kolls JK. *Immunology*. 2019 Jan;156(1):3-8.) whose elimination requires IL-17. Thus this observation in and of itself may have limited additional impact given that there are already several other publications involving *S. pyogenes* suggesting a similar phenomenon (referenced by the authors). However, understanding the mechanisms of IL-17 mediated protection of the respiratory tract is quite important and while the authors do provide some data regarding macrophages (see comments below) a key question is whether protection is mediated by local resident memory T cells within the mucosa, and if the PHAD containing vaccine more efficiently induces localization of T cells to NALT or the respiratory mucosa than vaccines lacking PHAD. One of the reported advantages of mucosal rather than parenteral immunization is the ability of mucosally administered vaccines to induce resident memory, and therefore understanding this issue in additional detail would contribute important mechanistic insights to the manuscript.

The role of T_{RM} cells in IL-17-mediated protection was assessed by flow cytometry in the lungs of vaccinated mice. Six-weeks post-last-immunisation lung digests were stained with a flow cytometry panel to assess T_{RM} cells CD45⁺CD3⁺CD4⁺CD62L⁺CD44⁺CD103⁺CD69⁺ cells (Figure 6). Although there was no change in the total fraction of T_{RM} cells within the pool of CD4 T-cells, we observed that J8-Lipo-DT-PHAD vaccination induced increased expression of CCR6 in T_{RM} cells (Figure 6F). As CCR6 is predominantly expressed by T helper type 17 (Th17) cells, this data suggests that J8-Lipo-DT-PHAD induces the production of IL-17 by T_{RM} cells.

The role of PHAD/DT in vaccination was assessed in IL-17A secretion (Figure 1G-H) and assessment of CCR6⁺ T_{RM} cells (Figure 6G). We observed increased secretion of IL-17A by immune cells isolated from lungs and spleen following *in-vitro* boosting with J8-DT only when all components of the vaccine 'J8-Lipo-DT-PHAD' were combined (Figure 1G-H). Absence of J8 (Lipo-DT-PHAD) or simultaneous removal of PHAD and DT specifically reduced the expression of CCR6⁺ on T_{RM} cells (Figure 6G). It is also important to note that there was no significant difference in Th17 T_{RM} cells between the J8-Lipo-DT-PHAD and J8-Lipo-DT vaccinated groups (Figure 6G). Assessment of Th17 T_{RM} was conducted six-weeks post last vaccine boost. There may have been a more distinct difference between the two vaccine cohorts if the study was conducted for over one year (as demonstrated with enduring antibody immunity in Figure 3B-E).

As expected, T_{RM} cells were not observed in lymphoid organs (spleen and NALT). Please refer to the below figure (Figure 2).

[FIGURE REDACTED]

Figure 1: T_{RM} cells in lymphoid organs.

2. *The authors should more clearly define the role of DT in this system and include experiments with vaccine lacking DT. The authors suggest that DT is providing “T cell help”, which may be important to for antibody production. However, given that the important end-point here appears to be a cell mediated response to the J8-peptide (rather than DT) the relevant helper response appears to be to J8 itself, leading to the question of whether DT is acting as a critical adjuvant independent to a role as a carrier protein providing help, or is in fact potentially not necessary.*

The role of DT in J8-Lipo-DT-PHAD-mediated immunity and protection was assessed in Figure 1E-F. Mice immunised with J8-Lipo-PHAD (without DT) had significantly reduced J8-specific salivary IgA responses in comparison to J8-Lipo-DT-PHAD (**p<0.001; Figure 1E). These mice also demonstrated significantly more bacterial burden in the NALT (**p<0.01) and lungs (#P<0.05) when compared to J8-Lipo-DT-PHAD immunized mice (Figure 1F). We also show that mice vaccinated with J8-Lipo-PHAD have significantly reduced IL-17A production in the lungs when compared to J8-Lipo-DT-PHAD (*p<0.05). This data suggests that DT is critical in the vaccine construct.

3. *The authors clearly demonstrate the PHAD enhances vaccine mediate protection, and the authors hypothesize that this is based on TLR4-mediated adjuvant effects, but this mechanism requires further evaluation. Certainly, demonstrating loss of efficacy in a TLR4 or Myd88 knockout mouse would add important mechanistic detail, as would further insights into whether TLR4 agonists enhance the generation of protective mucosal versus (potentially non-protective) systemic Th17 responses. Additional details regarding the cell-specific requirement for TLR4 signaling on different sets of APCs for effective mucosal vaccination could be novel and increase impact.*

Thank you for your suggestion. We have included this experiment in our future work and have listed this as a caveat in our discussion. Please refer to page 13, lines 416-418 of the manuscript.

4. *Clodronate-mediated macrophage depletion experiments are interesting but further evaluation of the underlying mechanisms would add to the manuscript. A key question is whether depletion of macrophages interferes with the IL-17 response itself or rather leave IL-17 responses intact, suggesting as the authors propose that macrophages are potential effectors of IL-17 mediated responses in this system. Identifying increased expression of IL-17 regulated genes such as Cxcl1 in macrophages isolated from vaccinated mice following challenge would further support a critical role for macrophages as effectors of the Th17 response.*

Thank you for your suggestion. We have now assessed whether depletion of macrophages interferes with the IL-17A response (Figure 7I). We have shown that macrophage depletion significantly reduces the IL-17A response in the lungs. This was not surprising as macrophages drive the differentiation of Th17 cells through IL-23. Our data looking into CCR6⁺ T_{RM} cells suggests that Th17 cells are the main source of IL-17A following i.n. vaccination with J8-Lipo-DT-PHAD (Figure 6F). Therefore, reduction of IL-17A following macrophage depletion further supports that macrophage and IL-17A work together as effectors in J8-Lipo-DT-PHAD-mediated immunity.

We will assess CxCl1 gene expression in our future work.

Reviewer #3 (minor comments)

1. *It is not clear what each dot represents in Figure 1. Does each dot represent the mean (or geomean) of one of 6 individual experiments? How many mice does each dot represent and how many mice in total are being evaluated?*

Figure 1 has been moved to the supplementary material as Supp. figure 1. Each dot represents the geomean of 10-15 mice. The figure is a summary of 6 separate experiments.

2. *The data in 2E is expressed as geomean +/- SD, but then significance is tested using Mann-Whitney. If data is non-parametric, median and interquartile range are more appropriate descriptive statistics.*

Figure 2E is now Figure 1D in the updated manuscript. To satisfy the non-parametric distribution of data, we have presented mice raw CFU data-points as a scatter-dot plot overlay on the geomean bar graph. With this method we are showing the whole data range in addition to geomean \pm geometric SD.

3. Figure 3 summarizes a large amount of data much of which is not clearly described in the text. For instance, there are several different mouse strains included without a clear rationale. Further, similar to the data in Figure 2, if using Mann-Whitney, mean and SD are not appropriate descriptive statistics. Also, exactly what “percent reduction” in these comparisons refers to and whether it is a statistically valid and/or requires confidence intervals is not well defined.

Figure 3 is now Figure 2 in the updated manuscript. More detail has now been included in the figure legend. Please refer to reviewer #2, comment 3 for a more detailed description of the statistics used in this manuscript.

4. *It would be helpful to describe methodology of boosting used for the data displayed in Figure 4.*

Figure 4 is now figure 3 in the updated manuscript. Vaccinated rested mice were boosted intranasally (*in-vivo*) on day 455 and antibody responses were assessed six days following boosting. This has now been clarified in the manuscript. Please refer to page 7, line 201-242 of the manuscript.

5. *In Figure 5 the authors demonstrate that protection from colonization following vaccination remains intact in mice lacking mucosal or total antibodies. This data would be enhanced by demonstrating the maintenance of both systemic and mucosal IL-17 responses to Ag rechallenge in these mice.*

Thank you for your suggestion. We have now shown that vaccinated plgR^{-/-} have significantly more IL-17A secretion in spleen cell supernatants in comparison to C57BL/6 (wild-type) mice (Figure 4F). We will take this suggestion on board and design future experiments to assess

systemic and mucosal IL-17A responses in pIgR^{-/-} and MuMT mice following Ag rechallenger in these mice.

6. Data in figure 6 would be stronger if the protective response to vaccination was directly compared in WT and IL-17-deficient mice.

As per reviewer #2, comment 5 – We do not think it is appropriate to compare WT and knock-out (KO) mice as there is always a chance that KO mice may have other compensating factors that may contribute to the experiment outcome. We believe that the appropriate comparison is between vaccinated and control KO mice as they only have one variable and that is their vaccination status. Comparing WT and KO mice may bring into play other genetic factors and introduces another variable into the experiment.

REVIEWER COMMENTS

Reviewer #1 (Remarks to the Author):

The authors have addressed a major concern by conducting neutrophil as well as macrophage depletion (new Fig 7) and demonstrated that BOTH neutrophil and macrophage depletion reduced the protective effect of the TLR4-agonist containing mucosal vaccine. These data are clearly presented and added to the Abstract and Discussion.

This manuscript provides significant and novel findings on T cell mediated protection following mucosal immunization against Streptococcal infection.

Reviewer #2 (Remarks to the Author):

The authors have undertaken a number of additional experiments to further delineate the protective mechanisms by which their vaccine might work, using knockout mice and antibody depletion methods to examine both humoral and cell mediated immune responses. The strengths of the paper are now very much in this analysis and showing that cell mediated effects and T cells might be very important in addition to antibodies.

The authors argue that, because no significant difference was seen between saline and adjuvant only (see figure 1 D and F), it was adequate to use saline as a control for all subsequent in vivo vaccine experiments, albeit that adjuvants have non specific enhancing effects on many aspects of the immune system. I can see it would not be feasible to repeat all the experiments with revised controls but this does mean that the wider benefits of PHAD might not be clear to the reader.

How effective is the vaccine ? It seems to clearly augment antibody responses but the bacteriology data are confusing. In Figure 1D, J8/Lipo-DT/PHAD outperforms saline and the Lipo-DT/PHAD adjuvant control, but it does not outperform J8/Lipo-DT i.e. the additional effect of PHAD is not apparent. The second experiment shown in Figure 1F suggests that PHAD does confer a benefit. Figure 2 then represents a series of different experiments that compare the vaccine with saline. Lots of experiments were done but why (rather than a

single adequately powered study)? In almost all cases there is a downward trend in the CFU. However looking at each of these: NALT, the vaccine was protective in 5/11 experiments; and in the lung, the vaccine was protective in 6/11 experiments. How to interpret this? The authors are commended on open-ness in this regard but in an ideal setting a single experiment with adequate power would have sufficed.

The percentage reductions and the summary graphs (panels BDFH) are distracting -these are log scale graphs and the reader will recognise the scale of reduction. The CFU graphs show CFU per organ but many of the data points are at 10^1 – was there a limit of detection for CFU which might impact calculation of percentage reduction? Unless the authors plated out the entire lung – this was not clear in the methods.

Figure 3 is now somewhat easier to follow. Here there seems little doubt that the addition of PHAD augments antibody response. With regard to the bacteriology however, is the vaccine J8/Lipo-DT/PHAD superior to J8/Lipo-DT and control, since many of the data points overlap ? The authors show the potential for these antibodies to opsonize by preincubating bacteria with them, and showing improved clearance in mice (J) but this might be construed to mean that passively donated antibodies are more important than a vaccine?

Minor point

The abstract still refers to mucosal infection. This should be changed to resp tract I think (or just infection?)

Reviewer #3 (Remarks to the Author):

The primary point of interest of this paper is the observation that the J8-lipo-DT-PHAD vaccine reduces the bacterial burden in the NALT and Lung following challenge with *S. pyogenes* and the protective effect depends on IL-17, CD4+ T cells, macrophages, neutrophils but not antibody.

The main issue with this manuscript in its current form is lack of clear experiments to determine the mechanistic factors that drive the Th17 response. This is best exemplified in

Figures 1F and 1G. In 1F it seems clear that J8-Lipo-DT-PHAD offers the best protection of all the vaccine configurations. 1G then attempts to define the association of protection with the IL-17 response in the lungs and is very confusing. The authors present data following ex vivo stimulation with J8-DT, DT, J8c, and *S. pyogenes*. A significant problem in interpreting this data is they do not show cytokine production by unstimulated cells. Therefore, we do not know if IL-17 levels are in fact due to response to the stimulation or constitutive. If, in fact, these are constitutive, then it appears that many of the vaccination preps have the ability to induce constitutive IL-17 production in the lung (presumably in a non-antigen-specific fashion), but it doesn't explain why J8-Lipo-DT-PHAD offers the best protection. If, on the other hand, the IL-17 levels are indeed induced by the ex-vivo stimulation, there are a number of issues that need to be addressed. While J8-DT induces higher levels of IL-17 secretion in mice vaccinated with J8-Lipo-DT-PHAD than in mice vaccinated with other vaccine configurations, J8c or DT alone do not, and appear to induce IL-17 in mice vaccinated with preparations that do not contain J8, raising the question of whether vaccine T cell responses are even specific to J8 or again may represent bystander responses, or potentially responses to DT, although DT alone does not induce a response above baseline either. Thus, it is very hard to understand how this vaccine is working and whether it is in fact inducing a J8-specific response, or inducing a non-specific response in the lung. Either of these would be interesting but this needs to be sorted through. One way to get at this would be to directly quantify J8 and DT specific Th17 cells in the lung and compare these to total Th17 cells in the lung using intracellular flow-cytometry following vaccination with different configurations.

Based on the data shown in Figure 6 the authors have attempted to begin investigating IL-17 producing T cells in the lungs of vaccinated mice, but there are several problems here. It is generally recognized that Trm are a subset of Tem, not Tcm, as suggested by the figure. CD4 Trm in the lung will most likely be both CD103 positive and negative so would not exclude those that are CD103 negative. As far as I know CCR6 expression has not been validated as a marker for Th17 cells in the mucosa (as opposed to the blood) and the authors should supply a reference and validate if they intend to use this marker. Even so, the difference in CCR6 expression following vaccination is trivial and these experiments would be much more informative if the authors used intracellular cytokine staining. Finally, these lung cells are

not stimulated ex vivo, therefore determining whether the vaccine induces J8 specific, DT-specific, and/or non-specific Th17 cells can't be determined, and as detailed above, this information is critical to understanding mechanism.

We thank the reviewers for their detailed analyses to help strengthen the findings in the manuscript. We have addressed the important points raised by the reviewers.

In addition to the changes in the manuscript outlined below for each Reviewer, we have changed:

- The title of the manuscript to reflect the role of neutrophils in vaccine-mediated protection.
- We have added a new figure panel to the manuscript (Figure 2). This figure includes – characterisation of lung Th17 cells from mice vaccinated with all possible vaccine configurations (Figure 2a-b) and assessment of secreted IL-17A from the lungs and spleens of these mice following *ex-vivo* stimulation with J8-DT, J8 or DT in (Figure 2c-d). Figure 2c-d were previously Figure 1g-h. This figure specifically addresses the questions raised by Reviewer #3.
- We have moved Figure 2 - ‘Summarized overview of multiple URT infections with J8-Lipo-DT-PHAD and J8-Lipo-DT vaccinated mice’ to Supp. Figure 3.
- We have amended Figure 3 to focus only on *in-vivo* boosting with *S. pyogenes* for the antibody (Figure 3f-i) and cytokine (Figure 3j-l) data. We have removed the *in-vivo* boosting with J8-DT antigen and focused on *S. pyogenes* boosting, which is relevant to the re-infection scenario in humans. We have changed the way we represent the cytokine data (Figure 3j-l). For the cytokine data, we have graphed PBS and J8-Lipo-DT-PHAD on the same graph to make the data easier to follow and represented IL-17A, IL-6 and IL-2 on separate graphs (Figure 3j-l). We have also moved the preincubation challenge experiment (previously Figure 3i-j) to Figure 4a-b.
- We have removed CCR6 from our T_{RM} analyses (as pointed out by Reviewer #3), re-analysed the data and now only present CD103⁺CD69⁺ T_{RM} cells from mice vaccinated with all possible vaccine configurations (Figure 7a-b). This was previously Figure 6.
- We have moved the macrophage and neutrophil depletion optimization data (previously Figure 7a-f) to Supp. Figure 8a-f. The protection and cytokine data (previously Figure 7g-i) are now Figure 8a-c.
- We have also included an additional author – Jacqueline Kaden.

All other amendments in the manuscript are summarized below.

Reviewer #1 (Remarks to the Author):

The authors have addressed a major concern by conducting neutrophil as well as macrophage depletion (new Fig 7) and demonstrated that BOTH neutrophil and macrophage depletion reduced the protective effect of the TLR4-agonist containing mucosal vaccine. These data are clearly presented and added to the Abstract and Discussion.

This manuscript provides significant and novel findings on T cell mediated protection following mucosal immunization against Streptococcal infection.

We thank Reviewer #1 for their constructive feedback for the manuscript.

Reviewer #2 (Remarks to the Author):

The authors have undertaken a number of additional experiments to further delineate the protective mechanisms by which their vaccine might work, using knockout mice and antibody depletion methods to examine both humoral and cell mediated immune responses. The strengths of the paper are now very much in this analysis and showing that cell mediated effects and T cells might be very important in addition to antibodies.

The authors argue that, because no significant difference was seen between saline and adjuvant only (see figure 1 D and F), it was adequate to use saline as a control for all subsequent in vivo vaccine experiments, albeit that adjuvants have non-specific enhancing effects on many aspects of the immune system. I can see it would not be feasible to repeat all the experiments with revised controls, but this does mean that the wider benefits of PHAD might not be clear to the reader.

We thank Reviewer #2 for their comments and providing us an opportunity to address this in the manuscript. We agree that adjuvants may have non-specific effects on many aspects of immune system and therefore to define specificity of vaccine mediated responses we have included all possible vaccine configurations (detailed below) for *in-vivo* and *in-vitro* assessments. We have now demonstrated the benefits of including PHAD in the vaccine in a range of readouts, as described below:

- **Enhanced mucosal immunity in J8-Lipo-DT-PHAD immunized mice.** We have compared the protective efficacy of J8-Lipo-DT-PHAD vaccine with vaccine construct without J8 (Lipo-DT-PHAD or Lipo-DT; Figures 1c-d), without DT (J8-Lipo-PHAD) or without PHAD (J8-Lipo; Figures 1e-f). These experiments were conducted in response to Reviewer #2's comment 4 in the first round of remarks - *"The authors examine mechanism of vaccine-induced protection in several ways (mice lacking antibodies, mice lacking IL17, CD4 depletion, etc). However, in each case, they have not undertaken the correct control group comparison. It is entirely possible that adjuvanted liposomes will confer greater protection than PBS alone - even in mice that lack antibodies. But it isn't clear that we are seeing a 'vaccine' effect here, or some other 'boosted' non-specific aspect of innate immunity. Surely the control comparison would have been with liposomes+DT"*. We demonstrate that mice immunized with J8-Lipo-DT-PHAD had significantly more salivary IgA (Figure 1c and e) and significantly better protection in comparison to PBS and the vaccine constructs that did not contain J8, DT or PHAD (Figure 1d and f). To further delineate the benefits of PHAD, in the 2nd round of revision we conducted a third set of efficacy experiments including J8-Lipo-DT-PHAD and various other controls (Figures 1g and h). This time we included the controls PBS, J8-Lipo-DT (as per previous two experiments) and additional two "delivery system" controls (Lipo-PHAD and Lipo 'liposomes alone'). Concordant with previous data, mice vaccinated with J8-Lipo-DT-PHAD had significantly higher J8-specific salivary IgA and IgG in comparison to all other vaccinated cohorts (Figure 1g). Following URT challenge with *S. pyogenes*, J8-lipo-DT-PHAD vaccinated mice had significantly reduced bacterial burden in the NALT in comparison to all other vaccinated cohorts (Figure 1h). The saline (PBS) group had comparable bacterial load to the other controls. These findings support the data outlined in Figures 1d and 1f. Together, in 2 out of 3 experiments we demonstrate that mice vaccinated with J8-Lipo-DT-PHAD have significantly better protection in comparison to mice immunized with J8-Lipo-DT (Figure 1f and 1h). Furthermore, we also show that DT is an integral component of this vaccine and works in synergy with J8, as its removal from a J8 containing construct ablates immunity (Figure 1f, vaccine construct 'J8-Lipo-PHAD').
- **Induction of IL-17A⁺CD4⁺ and T_{RM} cells in the lungs of J8-Lipo-DT-PHAD immunized mice.**
 - Following *ex-vivo* stimulation with J8-DT, mice immunized with J8-Lipo-DT-PHAD have significantly more ($P < 0.05-0.01$) IL-17A⁺CD4⁺ lung cells in comparison to mice immunized with PBS, Lipo-DT, Lipo-DT-PHAD, Lipo-PHAD, J8-Lipo-PHAD and Lipo (Figure 2b).
 - Following *ex-vivo* stimulation with J8-DT or J8, mice immunised with J8-Lipo-DT-PHAD have significantly more ($P < 0.05-0.0001$) IL-17A secretion in the lungs in comparison to PBS and all other vaccine configurations (including J8-Lipo-DT) and following J8-DT stimulation in the spleen (Figure 2d)
 - Mice immunized with J8-Lipo-DT-PHAD have significantly more ($P < 0.05$) T_{RM} cells in the lungs in comparison to PBS (Figure 7b).

Overall, in three separate adequately controlled experiments we have clearly shown that the bacterial burden in the saline (PBS) group is comparable to the mice vaccinated with incomplete vaccine or delivery system controls: Lipo-DT-PHAD or Lipo-DT (Figures 1c-d); J8-Lipo-PHAD or J8-Lipo (Figures 1e-f); Lipo-PHAD or Lipo (Figures 1g-h). These data demonstrate that PBS is an adequate negative control to be used in all future experiments. We also demonstrate that both J8 and DT along with PHAD are required for a complete and superior liposomal vaccine. Removal of either compromises vaccine's efficacy.

The results section for Figure 1 has been updated. Please refer to page 6, lines 173-189.

How effective is the vaccine? It seems to clearly augment antibody responses, but the bacteriology data are confusing. In Figure 1D, J8/Lipo-DT/PHAD outperforms saline and the Lipo-DT/PHAD adjuvant control, but it does not outperform J8/Lipo-DT i.e. the additional effect of PHAD is not apparent. The second experiment shown in Figure 1F suggests that PHAD does confer a benefit.

- As highlighted by Reviewer #2 -“J8-Lipo-DT-PHAD shows better protection when compared to J8-Lipo-DT. However, this protection only reached significance in one out of two experiments (Figures 1D and F)”. To confirm these data, we conducted a third efficacy experiment where J8-Lipo-DT-PHAD and J8-Lipo-DT were again compared head-to-head (Figure 1h). In this experiment we show that mice vaccinated with J8-Lipo-DT-PHAD have significantly reduced bacterial burden in the NALT and lungs when compared to the J8-Lipo-DT vaccinated group ($p < 0.05$). Overall, in two out of three experiments comparing all possible combinations and delivery system controls of the J8-Lipo-DT-PHAD construct, we show that the complete vaccine “J8-Lipo-DT-PHAD” is superior in immunogenicity and protective efficacy to J8-Lipo-DT.
- We have previously demonstrated the efficacy of J8-Lipo-DT vaccine against *S. pyogenes* (PMID: 27976706). In the current manuscript we demonstrate that the addition of PHAD to the J8-Lipo-DT vaccine leads to a significantly higher and longer sustained antibody responses when compared to the J8-Lipo-DT vaccine alone (Figure 3f-i). Utilizing a pre-incubation challenge assay, we further demonstrate significantly enhanced functionality of J8-Lipo-DT-PHAD vaccine antibodies in comparison to J8-Lipo-DT (Figure 4a-b). Inclusion of PHAD into the J8-Lipo-DT vaccine also leads to enhanced T-cell/IL-17A responses in comparison to J8-Lipo-DT vaccine (Figure 2). Lung cells from mice vaccinated with J8-Lipo-DT-PHAD and stimulated *ex-vivo* with J8-DT or J8 show significantly more ($P < 0.05$ - $P < 0.0001$) IL-17A secretion in comparison to mice vaccinated with J8-Lipo-DT and all other controls (Figure 2c). Additionally, lung cells from mice vaccinated with J8-Lipo-DT-PHAD have significantly more CD4⁺ tissue resident memory (CD103⁺CD69⁺T_{RM}) cells in comparison to PBS ($P < 0.01$; Figure 7b).

Together, these data demonstrate that inclusion of PHAD in the vaccine enhances the effector function of B- and T-cells and offers benefits in comparison to vaccine constructs without PHAD.

Figure 2 then represents a series of different experiments that compare the vaccine with saline. Lots of experiments were done but why (rather than a single adequately powered study)? In almost all cases there is a downward trend in the CFU. However, looking at each of these: NALT, the vaccine was protective in 5/11 experiments; and in the lung, the vaccine was protective in 6/11 experiments. How to interpret this? The authors are commended on open-ness in this regard but in an ideal setting a single experiment with adequate power would have sufficed.

- The data in Figure 2 (now Supp. Figure 3) are representative of various experiments (over a period of 5 years) conducted with J8-Lipo-DT-PHAD and PBS (Supp. Figure 3a-d) or J8-Lipo-DT and PBS (Supp. Figure 3e-h). These experiments were not conducted solely for the purpose of this figure. The data in Supp. Figure 3 have been extracted from other experiments (some presented in other figures in the current manuscript, as highlighted in the figure legend) to show the inherent variability of the URT challenge model. In Supp. Figures 3b and 3d, we have combined the data from 115 J8-Lipo-DT-PHAD vaccinated mice and 115 PBS vaccinated mice and demonstrated that J8-Lipo-DT-PHAD vaccinated mice have significantly reduced ($P < 0.0001$; $>90\%$ reduction) bacterial burden in comparison to PBS vaccinated mice.

All experiments in this manuscript are adequately powered. For all challenge experiments we typically use 10 mice/group as determined by power calculations. For power calculations a Shapiro-Wilk test was used to calculate that the data was not normally distributed. The parent distribution input parameter for the power analysis was logistic due to the heavy tails on the distribution. Based on data from 5 separate challenge experiments and the assumptions (listed below), an average group number of **10** will provide a medium effect size and a power of 0.8.

For our statistical analysis, our assumptions were as below;

- Power of 0.8.
 - Alpha value of 0.05.
 - Two sided (To test **both** if the mean is significantly greater than x and if the mean significantly less than x).
 - Not normally distributed.
- In addition, due to logistical reasons, we only ever recommend a maximum experiment cohort of 40 mice in total. URT challenge experiments are labour-intensive and time-consuming experiments that always require at least two skilled personal to perform the hand-on procedures. There is a likelihood of introducing variability into the experiment with increasing the cohort size. Firstly, it is critical that the challenge inoculum we use is consistent from mouse 1 to mouse 40. The longer the inoculum sits on ice and is removed/put back on ice the higher the chance that bacterial viability will be affected. This will lead to inconsistent challenge dose between mouse 1 and 40. Secondly, these mice are followed for 2-3 days and on each day, mice are weighed, clinically scored (2x a day) and throat swabs and nasal shedding collected to assess/monitor the infection. On the final day, mice are euthanized, tissues collected, homogenized and plated out onto agar plates for bacterial burden enumeration. These are all long procedures. In Figure 1, we pushed the 40 mice experiment limit to 50 mice/experiment. We do not think it will be possible to repeat these experiments to include all the controls within the one experiment. This will be a minimum 70 mice experiment with the following groups - PBS, J8-Lipo-DT-PHAD, J8-Lipo-DT, Lipo-DT-PHAD, Lipo-DT, J8-Lipo-PHAD, J8-Lipo, Lipo-PHAD and Lipo. The next best thing to do would be to split this up into three main experiments, which we have now done (please refer to Figure 1). The validity of these experiments is confirmed by including common controls in each of the three experiments. These controls are PBS, J8-Lipo-DT-PHAD and J8-Lipo-DT.

Additionally, for the last 10 years we have been continuously optimizing and improving the URT challenge model which closely mimics the URT infection in humans. We do agree that some of the data are variable, and although there is several fold reduction in the CFU counts in the group of interest, it does not reach statistical significance in all experiments. However, we do believe that these data are highly relevant, as this is the only murine model available currently to assess vaccine-mediated immunity and protection following *S. pyogenes* URT challenge.

The percentage reductions and the summary graphs (panels BDFH) are distracting -these are log scale graphs, and the reader will recognise the scale of reduction. The CFU graphs show CFU per organ but many of the data points are at 10^1 – was there a limit of detection for CFU which might impact calculation of percentage reduction? Unless the authors plated out the entire lung – this was not clear in the methods.

The main purpose of Supp. Figure 3 is to highlight the variability of the URT challenge model (as described above). Even though we do not see a significant reduction in bacterial load between the vaccinated and saline (PBS) cohorts in every experiment, when we combine the data into one figure (Supp. Figures 3b, d, f and h), there is a >90% reduction in bacterial load in the vaccinated cohort when compared to the saline cohort, which is also significant. These summary panels show the magnitude of protection associated with vaccination in comparison to saline alone. Furthermore, the reason as to why we show % reduction on top of the CFU graph is to emphasize, in some cases, that despite the data not being statistically significant, the magnitude of reduction in CFU in the vaccinated group in comparison to the PBS group is noteworthy. It will be impossible to conduct an experiment with >100 mice/group to show the same level of protection presented in Supp. Figure 3.

To determine bacterial burden in tissues, we harvest the whole tissue, homogenize it and perform 10-fold serial dilutions from neat to 10^6 . We then plate out 20 μ L dots for each dilution in duplicate onto CBA plates. Our limit of detection is to count between 1-30 CFU/dot. In samples where we only see 1-2 CFU in the neat dot, we re-plate the neat sample in a 100 μ L volume as a spread plate to confirm the low counts.

Figure 3 is now somewhat easier to follow. Here there seems little doubt that the addition of PHAD augments antibody response. With regard to the bacteriology however, is the vaccine J8/Lipo-DT/PHAD superior to J8/Lipo-DT and control, since many of the data points overlap? The authors show the potential for these antibodies to opsonize by preincubating bacteria with them, and showing improved clearance in mice (J) but this might be construed to mean that passively donated antibodies are more important than a vaccine?

The bacteriology data in Figure 3e (previously Figure 3h), show that mice vaccinated with J8-Lipo-DT-PHAD or J8-Lipo-DT and rested for over 200 days are significantly protected in comparison to mice vaccinated with PBS. This experiment was terminated on day 1 post-challenge due to 60% of the PBS cohort presenting with high clinical scores, therefore we believe that the vaccine did not have sufficient time to exert its full effect and the difference between the two-vaccine groups (J8-Lipo-DT-PHAD and J8-Lipo-DT) - 56% reduction in the NALT and 99% reduction in the lungs - did not reach statistical significance.

The pre-incubation challenge experiment in Figure 4b illustrates the functionality of vaccine-specific antibodies. We show that neat serum from J8-Lipo-DT-PHAD mice is opsonic, as pre-incubation of *S. pyogenes* with this serum is able to elicit a downstream protective response following challenge. The purpose of this experiment was to explicitly demonstrate the functionality of vaccine antibody in opsonophagocytic killing of *S. pyogenes*. We have not tested the protective efficacy of passively donated antibodies (i.e., intravenous administration of serum/purified antibody pre-or post-challenge), therefore we cannot say that passively donated antibodies are more important than a vaccine. However, we can say that vaccine antibodies, even at low concentrations in the serum, are functional.

Minor point

The abstract still refers to mucosal infection. This should be changed to resp tract I think (or just infection?)

This has now been changed to 'respiratory tract infection'. Please refer to page 1, line 25 of the manuscript.

Reviewer #3 (Remarks to the Author):

*The primary point of interest of this paper is the observation that the J8-lipo-DT-PHAD vaccine reduces the bacterial burden in the NALT and Lung following challenge with *S. pyogenes* and the protective effect depends on IL-17, CD4⁺ T cells, macrophages, neutrophils but not antibody.*

We would like to thank Reviewer #3 for their constructive feedback and suggestions on how to improve the manuscript.

- J8-Lipo-DT-PHAD is able to induce both cellular and humoral immune responses. The data in the manuscript clearly demonstrates the importance of cellular immunity over humoral immunity in the mucosal compartment. Any role that mucosal antibody plays would be dependent on CD4⁺ T-cells, which would function as helper cells and activators of macrophages and neutrophils through IL-17A.

*The main issue with this manuscript in its current form is lack of clear experiments to determine the mechanistic factors that drive the Th17 response. This is best exemplified in Figures 1F and 1G. In 1F it seems clear that J8-Lipo-DT-PHAD offers the best protection of all the vaccine configurations. 1G then attempts to define the association of protection with the IL-17 response in the lungs and is very confusing. The authors present data following ex vivo stimulation with J8-DT, DT, J8c, and *S. pyogenes*. A significant problem in interpreting this data is they do not show cytokine production by unstimulated cells. Therefore, we do not know if IL-17 levels are in fact due to response to the stimulation or constitutive. If, in fact, these are constitutive, then it appears that many of the vaccination preps have the ability to induce constitutive IL-17 production in the lung (presumably in a non-antigen-specific fashion), but it doesn't explain why J8-Lipo-DT-PHAD offers the best protection. If, on the other hand, the IL-17 levels are indeed induced by the ex-vivo stimulation, there are a number of issues that need to be addressed. While J8-DT induces higher levels of IL-17 secretion in mice vaccinated with J8-Lipo-DT-PHAD than in mice vaccinated with other vaccine configurations, J8c or DT alone do not, and appear to induce IL-17 in mice vaccinated with preparations that do not contain J8, raising the question of whether vaccine T cell responses are even specific to J8 or again may represent bystander responses, or potentially responses to DT, although DT alone does not induce a response above baseline either.*

Please note that due to major revisions in the manuscript, the figure numbers have changed. The previous Figure 1g is now Figure 2c.

We apologize for not showing the cytokine data from unstimulated (media) cells. We have now included this in the figure. Please refer to Figure 2c and pages 6-7, lines 191-209 of the manuscript.

- In Figure 2c we have normalized the data to the media (unstimulated) control and represented the data as mean \pm SEM. The re-analysis of the data shows that ex-vivo stimulation with J8-DT is able to induce IL-17A levels that are significantly higher ($P < 0.05$ - 0.0001) in J8-Lipo-DT-PHAD immunized mice in comparison to PBS and all other vaccine configurations (including J8-Lipo-DT; vaccine without PHAD) tested. Therefore, suggesting that IL-17A levels are indeed

vaccine-specific and not constitutive. The IL-17A data following stimulation with either J8 or DT in the lungs (Figure 2c) is also significantly higher for the J8-Lipo-DT-PHAD vaccinated cohort in comparison to PBS and all other vaccine configurations tested. J8 alone was able to stimulate an IL-17A response in mice vaccinated with Lipo-DT-PHAD (vaccine without J8) but this was significantly less ($P < 0.01$) than the response elicited by J8 from J8-Lipo-DT-PHAD vaccinated mice. The cross-reactivity between J8 and DT could be due to known sequence similarities between these two antigens (PMID: 12721940).

Thus, it is very hard to understand how this vaccine is working and whether it is in fact inducing a J8-specific response, or inducing a non-specific response in the lung. Either of these would be interesting but this needs to be sorted through. One way to get at this would be to directly quantify J8 and DT specific Th17 cells in the lung and compare these to total Th17 cells in the lung using intracellular flow-cytometry following vaccination with different configurations.

Thank you for your suggestion. We have now conducted intracellular flow-cytometry (ICS) to directly quantify J8 and DT specific Th17 cells in lungs and the data are presented in Figure 2b and on page 6, lines 191-209. The data demonstrate that both J8 and DT are required to stimulating Th17 cells and either antigen alone is unable to elicit this response. The results are summarized below:

- To directly quantify antigen-specific Th17 (IL-17A⁺CD4⁺) cells in lungs, BALB/c mice were vaccinated with PBS, J8-Lipo-DT-PHAD, J8-Lipo-DT, Lipo-DT, Lipo-DT-PHAD, J8-Lipo-PHAD, J8-Lipo, Lipo-PHAD or Lipo. Six-weeks following their last vaccine boost, mice were euthanized, lungs excised, and cells stimulated with J8-DT, J8, DT or left unstimulated. The percentage of antigen-specific Th17 cells was calculated in comparison to the total Th17 cells (cells stimulated with media) in the lungs and data are presented as boxplots with min/max (Figure 2b).
- Lung cells from mice vaccinated with J8-Lipo-DT-PHAD and stimulated with J8-DT had significantly more ($P < 0.05$) Th17 cells in comparison to lung cells from PBS, Lipo-DT, Lipo-DT-PHAD, Lipo-PHAD, J8-Lipo-PHAD and Lipo immunized mice, as determined by ICS (Figure 2b). Lung cells that were stimulated *ex-vivo* with J8 or DT did not elicit a Th17 response (data not shown). However, significant levels of secreted IL-17A were detected in supernatants of lung cells from J8-Lipo-DT-PHAD mice stimulated *ex-vivo* with J8-DT, J8 or DT (Figure 2c).

*Based in the data shown in Figure 6 the authors have attempted to begin investigating IL-17 producing T cells in the lungs of vaccinated mice, but there are several problems here. It is generally recognized that Trm are a subset of Tem, not Tcm, as suggested by the Figure 6. CD4 Trm in the lung will most likely be both CD103 positive and negative so would not exclude those that are CD103 negative. As far as I know CCR6 expression has not been validated as a marker for Th17 cells in the mucosa (as opposed to the blood) and the authors should supply a reference and validate if they intend to use this marker. Even so, the difference in CCR6 expression following vaccination is trivial and these experiments would be much more informative if the authors used intracellular cytokine staining. Finally, these lung cells are not stimulated *ex vivo*, therefore determining whether the vaccine induces J8 specific, DT-specific, and/or non-specific Th17 cells can't be determined, and as detailed above, this information is critical to understanding mechanism.*

Thank you for your critical analysis of Figure 6. We have now re-analysed the T_{RM} data to reflect your suggestions. We have removed CCR6 as a marker for Th17 cells. Please refer to Figure 7 (previously Figure 6) and page 11, lines 351-366. Please see below a breakdown of the results:

- As suggested by Reviewer #3, we re-analyzed the T_{RM} dataset from the previous version of the manuscript. This time we assessed T_{RM} (CD103⁻CD69⁺) cells in the T_{EM} (CD62L⁻CD44⁺) population instead of the T_{CM} (CD62L⁺CD44⁺) population. CD69 is a key marker that distinguishes memory T-cells in tissue in comparison to those that are in circulation and CD103 is predominantly expressed on CD8⁺TRM⁺ cells. Our data showed that mice vaccinated with J8-Lipo-DT-PHAD have significantly more T_{RM} in the lungs in comparison to PBS, J8-Lipo, J8-Lipo-PHAD and Lipo vaccinated cohorts (Figure 7b).
- We have performed *ex-vivo* antigen stimulation of lung cells and demonstrated the specificity of the Th17 of response, as detailed in Figure 2a-b and on pages 6-7, lines 191-209.
- As for determining T_{RM}⁺IL-17A⁺ cells, we do not think *ex-vivo* stimulation/ICS staining is the correct method. CD69 is an early activation marker (Gonzalez-Amaro, Trends Mol Med., 2013 - DOI: 10.1016/j.molmed.2013.07.006) that appears on the plasma membrane of activated cells; thus, results obtained from an ICS experiment will not reflect 'bona fide' T_{RM} cells. Th17 cells also lose their expression of their signature cytokine becoming "ex-Th17" cells when they mature into long-lived memory cells (Amezcuca Vesely, Cell, 2019 - DOI: <https://doi.org/10.1016/j.cell.2019.07.032>). Further studies looking into MHC tetramers and IL-17A tracking mouse models are required to delineate the antigen-specificity and origin of vaccine-induced T_{RM}⁺IL-17A⁺ cells in the lungs. This is beyond the scope of the current study and will be implemented in future studies. These data have been discussed in-depth on page 13-14, lines 436-459.

We have further expanded on the methodologies section to include TRM and ICS protocols. Please refer to page 19, lines 621-636.

We believe that we have fully justified the remarks of Reviewer #1, #2 and #3.

REVIEWER COMMENTS

Reviewer #2 (Remarks to the Author):

The flow of the manuscript is improved. Figure 1 includes three experiments with each of the vehicle controls included in at least one, and a control group included in all three. The immunological data in Figure 2 now include data from mice vaccinated with each control as well. By moving some figures to the supplementary, the paper is easier to follow though rather long.

Results

Line 123. the authors claim "Bacterial enumeration from NS and TS are qualitative in nature and were found to be highly variable". Others have undertaken both measurements, and the authors themselves have quantified these in Supp figure 1 though not in an easily understandable way. If NS and TS are to be shown in Supp figure 1 please explain the CFU counts- are these per swab? For nasal shedding is this per mouse per test or something else?

Same figure, please explain in the legend if the data are CFU per whole organ?

Does each data point represent the mean data from one experiment (n=10-15 mice)?

Lines 149-151, The sentences reporting on significance of antibody titers induced by a specific vaccine antigen in comparison to a control that doesn't contain the vaccine antigen seem overly long. Suggest condense.

Line 211-227. "An overview of experiments..." This section should be condensed. Supp fig 3: The implication remains that these experiments were repeated many times and this is not explained. Panels B and D remain confusing as presented.

Line 229- longevity of response. Better to clarify the two cohort design on both the diagram and also in the text, so that the help the reader navigate. The authors first of all introduce the idea of a study lasting 454 days but immediately go on to describe a challenge expt that ends on day 252...

The clinical scoring system could perhaps have been mentioned in the model development section-given that scores are provided in the results, and the study was terminated prematurely. The scoring system is mentioned in the methods but without detail of what the numbers mean or a reference.

Minor issues

Can improve the conciseness in several places

-Abstract – suggest condense and combine the two sentences: “.....was able to induce both cellular and humoral immunity. We demonstrate long-lived humoral and cellular memory responses following vaccination.”

-Introduction first paragraph – stats about mortality and burden of disease are repeated

Introduction second para lines 48-49. “Firstly, the pathophysiology of ARF and RHD is driven by autoimmune molecular mimicry between the *S. pyogenes* M-protein and carbohydrate cross-reacting with human-tissue.” This statement feels somewhat didactic- is there now proof?

-Line 117 “An overview of *S. pyogenes* (2031) URT infection in mice”. Suggest ‘Model development’ is more appropriate to help the reader understand what this paragraph is about

Typo line 134 tract

Line 243 “lower clinical scores” (not less?)

Reviewer #3 (Remarks to the Author):

1. As requested, the authors reanalyzed data regarding ex vivo stimulation of lung and spleen cells from animals immunized with different vaccine preps, and show that restimulation with J8 induces stronger IL-17 secretion in mice immunized with J8-Lipo-DT-PHAD than other preps. This is reassuring. It remains unclear why J8 induces a weaker but significant response in mice immunized with Lipo-DT-PHAD. This suggests some type of non-specific response to the vaccine preparation that could be important to protection and could be explored in follow-up studies. In this reanalysis the authors state that they have normalized values to unstimulated controls. This normalization method should be described in the figure legend or methods in further detail.

2. New experiments (Fig.2a) using ICS demonstrate significantly higher numbers of IL17+CD4+ in the lungs of mice immunized with J8-Lipo-DT-PHAD than other vaccine preps following ex vivo stimulation with J8-DT. I appreciate the authors candor that J8 of DT alone did not elicit an IL-17 response by ICS (although they did when IL-17 was measured by ELISA). However, this does leave some uncertainty in these experimental results regarding whether IL-17 responses are truly-J8 specific, although the evidence points in that direction.

3. The data shown in the new Figure 7 remains quite problematic.

a. The staining of lung cells for CD44 and CD62L shown in 7a is very hard to understand. The majority of T cells at mucosal surfaces including the lung are memory T cells and therefore would typically be CD62L-CD44+, while a distinct minor population of naïve T cells that stain CD62L+CD44- is also typically easily identified. The majority of the cells shown on the flow plot in 7a are either CD62L-CD44- (which is not a well described staining pattern) or CD62L+CD44+, suggesting Tcm, but these are typically relatively uncommon in the lung. Thus, the absence of distinct CD62L+CD44- Tnaive and CD62L-CD44+ Tm populations is difficult to explain. A nice example of T cell staining in the respiratory tract is shown in Curham LM, PMID: 36681765. These discrepancies should be resolved prior to including this data.

b. Once resolved, I would recommend not excluding CD4+CD103+CD69+ cells from the

analysis, but rather either not discriminating CD69+CD4 Trm based on CD103 (as there are likely both CD103+ and CD103- CD4 Trm in the lung), or showing both CD103+ and CD103- CD4 Trm.

Reviewer #2 (Remarks to the Author):

The flow of the manuscript is improved. Figure 1 includes three experiments with each of the vehicle controls included in at least one, and a control group included in all three. The immunological data in Figure 2 now include data from mice vaccinated with each control as well. By moving some figures to the supplementary, the paper is easier to follow though rather long.

Results

Line 123. the authors claim “Bacterial enumeration from NS and TS are qualitative in nature and were found to be highly variable”. Others have undertaken both measurements, and the authors themselves have quantified these in Supp figure 1 though not in an easily understandable way. If NS and TS are to be shown in Supp figure 1 please explain the CFU counts- are these per swab? For nasal shedding is this per mouse per test or something else?

We thank Reviewer #2 for their time and input into the manuscript. We have addressed all their comments in the section below.

Nasal shedding (NS) from mice were collected by pressing their nares (each press = 5 seconds) on to half a Columbia blood agar (CBA+5% defibrinated horse blood) plate five times. This was repeated for another five times on the second half of the agar plate. The five dots (created by pressing the nares onto the agar plate) were spread with a 10 µL inoculating loop and the plate was incubated overnight at 37°C. We used one agar plate/mouse/time-point. The next day the agar plate was removed from the incubator and cfus were counted per half of the plate and the data are represented as geometric mean \pm geometric SD (*S. pyogenes* shed from the nasopharynx).

Throat swabs (TS) were enumerated by swabbing mice throats (10 total rotations/swab) with a floq swab (Interpath; 516CS01) and then resuspending the swab in 250 µL of PBS for 30 seconds (with vigorous twisting). This suspensions (PBS+*S. pyogenes*) was diluted 1:10 until dilution 10^{-6} (6 x dilutions). Each dilution was dot plated in duplicate on to a CBA plate. The plates were incubated overnight at 37°C. Each dilution was counted (reliable counts between 10-40 dots) and the data are represented as cfu/mL of swab.

Each circle symbol (total 6 circles) presented in Supp. Fig. 1 is the geometric mean for the PBS only immunised (n=10-15) mice from one experiment. In total there are 6 circles for the 6 experiments presented in this figure.

The figure legend has been updated. **Please refer to page 44, lines 1639-1655.**

Same figure, please explain in the legend if the data are CFU per whole organ?

The data are represented as cfu/whole organ. Tissues were collected (NALT in 250 µL of PBS and lungs in 500 µL of PBS), homogenised and diluted 1:10 until dilution 10^{-6} (6 x dilutions). Each dilution was dot plated in duplicate on to a CBA plate. The plates were incubated overnight at 37°C. Each dilution was counted (reliable counts between 10-40 dots) and the data were represented as cfu/tissue.

The figure legend has been updated. **Please refer to page 44, lines 1639-1655.**

Does each data point represent the mean data from one experiment (n=10-15 mice)?

Each dot represents the geomean from one experiment. The figure legend has been updated. **Please refer to page 44 and lines 1640-1641**

Lines 149-151, The sentences reporting on significance of antibody titers induced by a specific vaccine antigen in comparison to a control that doesn't contain the vaccine antigen seem overly long. Suggest condense.

This sentence has been condensed. **Please refer to page 5, lines 146-148.**

Line 211-227. "An overview of experiments..." This section should be condensed. Supp fig 3: The implication remains that these experiments were repeated many times and this is not explained. Panels B and D remain confusing as presented.

The section "An overview of immunity across multiple experiments" has been moved to the supplementary section of the manuscript. **Please refer to page 46, lines 1691-1708.**

Panels b, d, f and h represent the combined percent reduction in CFU of the J8-Lipo-DT-PHAD mice in comparison to the PBS mice. Please see formula below:

% reduction formula = ((geomean cfu of PBS group – geomean cfu of J8-Lipo-DT-PHAD group)/geomean cfu of PBS group)*100. The formula has been included into the supplementary data. **Please refer to page 46, lines 1706-1708.**

Each symbol (square/circle) represents the average from one experiment. Panels b and d show 11 symbols (average of 11 x experiments) and panels e and g show 4 symbols (average of 4 x experiments).

Line 229- longevity of response. Better to clarify the two cohort design on both the diagram and also in the text, so that the help the reader navigate. The authors first of all introduce the idea of a study lasting 454 days but immediately go on to describe a challenge expt that ends on day 252...

This section of the manuscript has been amended to follow the chronological order of Figure 3. **Please refer to pages 7-8, lines 208-258.** Figure 3a has been replaced to represent the two-cohort design of the experiment. **Please refer to page 34.**

The clinical scoring system could perhaps have been mentioned in the model development section- given that scores are provided in the results, and the study was terminated prematurely. The scoring system is mentioned in the methods but without detail of what the numbers mean or a reference.

The clinical scoring system has been defined on **page 7, lines 221-225.** A reference has also been provided for the scoring system (Pandey & Good, 2020) in the methods on **page 18, lines 599-601.**

Minor issues. Can improve the conciseness in several places

-Abstract – suggest condense and combine the two sentences: ".....was able to induce both cellular and humoral immunity. We demonstrate long-lived humoral and cellular memory responses following vaccination."

The two sentences have been combined. Please refer to page 1, lines 19-22.

-Introduction first paragraph – stats about mortality and burden of disease are repeated

The introduction has been amended to remove the repeated statements. Please refer to page 2, lines 38-40.

Introduction second para lines 48-49. “Firstly, the pathophysiology of ARF and RHD is driven by auto-immune molecular mimicry between the *S. pyogenes* M-protein and carbohydrate cross-reacting with human-tissue.” This statement feels somewhat didactic- is there now proof?

Infection with *S. pyogenes* is associated with acute rheumatic fever (ARF), rheumatic heart disease (RHD) and post-streptococcal glomerulonephritis (PSGN). ARF, RHD and PSGN develop due to auto-antibodies and cellular responses that target heart, brain and kidney antigens. The sharing of host and streptococcal antigens (M-protein and group A carbohydrate) lead to these disease outcomes. Cross-reactive antigens of group A streptococcus and their sequellae have been reviewed by Madelene Cunningham (Cunningham, 2019).

-Line 117 “An overview of *S. pyogenes* (2031) URT infection in mice”. Suggest ‘Model development’ is more appropriate to help the reader understand what this paragraph is about

Thank you for your suggestion. This is an already established model that we developed and reported in three previously published manuscripts - (Ozberk et al., 2021; Zaman et al., 2021; Zaman et al., 2016). Therefore, we don't think changing the title of this section to 'Model development' is appropriate. The purpose of this section is to provide an overview of the URT infection model with the J8-Lipo-DT-PHAD and J8-Lipo-DT vaccines.

Typo line 134 tract

The typo has been fixed – trat to tract. Please refer to page 4, line 132.

Line 243 “lower clinical scores” (not less?)

“Lower” has been replaced with “less”. Please refer to page 7, line 219.

Reviewer #3 (Remarks to the Author):

1. As requested, the authors reanalyzed data regarding ex vivo stimulation of lung and spleen cells from animals immunized with different vaccine preps, and show that restimulation with J8 induces stronger IL-17 secretion in mice immunized with J8-Lipo-DT-PHAD than other preps. This is reassuring. It remains unclear why J8 induces a weaker but significant response in mice immunized with Lipo-DT-PHAD. This suggests some type of non-specific response to the vaccine preparation that could be important to protection and could be explored in follow-up studies. In this reanalysis the authors state that they have normalized values to unstimulated controls. This normalization method should be described in the figure legend or methods in further detail.

We thank Reviewer #3 for their valuable input and comments. We have addressed all their comments in the section below.

On page 7, lines 204-206 of the manuscript we state, “We also noted that Lipo-DT-PHAD vaccinated mice also responded to J8 (Figure 2c), which may be due to the known sequence similarities between J8 and DT (Batzloff et al., 2003)”.

For Figure 2b the % of IL-17A⁺ cells from lung cells that were stimulated with media were subtracted from the % of IL-17A⁺ cells from lung cells that were stimulated with J8-DT. This was conducted for each experimental group. For example – lung cells from J8-Lipo-DT-PHAD immunised mice were seeded out into two wells, the first well was stimulated with media and the second well was stimulated with J8-DT. Each well was assessed for % of IL-17A⁺ CD4⁺ cells. The % of cells from the media stimulated group was subtracted from the J8-DT stimulated group (to remove background).

The same method was applied for Figures 2c and d. However, in this case, instead of using percentages, secreted IL-17A in pg/mL were used. Therefore, secreted IL-17A (pg/mL) from media stimulated cells were subtracted from cells stimulated with J8-DT, J8 or DT.

The method for normalisation has been clarified in the figure legend. **Please refer to page 33, lines 1186-1189.**

2. New experiments (Fig.2a) using ICS demonstrate significantly higher numbers of IL17+CD4+ in the lungs of mice immunized with J8-Lipo-DT-PHAD than other vaccine preps following ex vivo stimulation with J8-DT. I appreciate the authors candor that J8 of DT alone did not elicit an IL-17 response by ICS (although they did when IL-17 was measured by ELISA). However, this does leave some uncertainty in these experimental results regarding whether IL-17 responses are truly-J8 specific, although the evidence points in that direction.

Thanks for raising this point. We believe that part of this discordance is because the data originated from ICS and cytokine detection assay in culture supernatant capture distinct phases of immune cell activation. In the ICS assay, we have detected IL-17 production in a narrow window of time (specifically between 16 and 21h post-initial activation, when brefeldin was added) while in CBA, we have measured overall accumulation of IL-17 in the culture supernatant during the first 72 hours of activation. Therefore, responses elicited by J8 and DT alone may be delayed in relation to J8-DT and were not captured by ICS. The second point can be related with sensitivity of the assay, as ICS may have not been able to detect small variations in IL-17A response following activation with J8 and DT alone. The third (and less likely) possibility is that both assays are capturing response of distinct immune cell types following activation. While the ICS data is specifically demonstrating Th17 responses, IL-17A captured by CBA can be originated from other immune cell types known to secrete IL-17A, including CD8⁺, gamma-delta T, invariant NKT and innate lymphoid cells.

3. The data shown in the new Figure 7 remains quite problematic.

a. The staining of lung cells for CD44 and CD62L shown in 7a is very hard to understand. The majority of T cells at mucosal surfaces including the lung are memory T cells and therefore would typically be CD62L-CD44+, while a distinct minor population of naïve T cells that stain CD62L+CD44- is also typically easily identified. The majority of the cells shown on the flow plot in 7a are either CD62L-CD44- (which is not a well described staining pattern) or CD62L+CD44+, suggesting Tcm, but these are typically relatively uncommon in the lung. Thus, the absence of distinct CD62L+CD44- Tnaive

and CD62L-CD44⁺ T_{RM} populations is difficult to explain. A nice example of T cell staining in the respiratory tract is shown in Curham LM, PMID: 36681765. These discrepancies should be resolved prior to including this data.

Thank you so much for providing us with further guidance on the analysis of immune cells in the lung. To identify the origin of this unusual CD62L⁺CD44⁺ population, we revisited the data from the flow cytometry experiment conducted to assess the impact of multiple vaccine constructs on T_{RM} responses. Surprisingly, when back-gating from the central memory plot, we observed that this unusual CD62L⁺CD44⁺ population sits on the gate of high side scatter (SSC) cells, not coinciding with SSC profile of lymphocyte populations. Therefore, we have contracted our initial FSC versus SSC gating to only capture the phenotype of cells within the lymphocyte gate and reanalysed our data. We have updated **Figure 7 (page 41)** of the manuscript and generated a new **Supp. Fig. 7 (page 54)** to show the final gating strategy. Regardless of the FSC vs SSC gating utilised, our data show that the population of CD69⁺CD103⁻ CD4⁺T_{RM} cells is increased in lungs of mice immunized with J8-Lipo-DT-PHAD. Consistent with findings reported by (Curham et al., 2023) the majority of CD4⁺T_{RM} identified in mice lungs were characterised as CD44⁺CD62L⁻CD103⁻CD69⁺. **Please refer to page 11, lines 348-356 and pages 13-14, lines 431-439.**

b. Once resolved, I would recommend not excluding CD4⁺CD103⁺CD69⁺ cells from the analysis, but rather either not discriminating CD69⁺CD4⁺ T_{RM} based on CD103 (as there are likely both CD103⁺ and CD103⁻ CD4⁺ T_{RM} in the lung), or showing both CD103⁺ and CD103⁻ CD4⁺ T_{RM}.

Following your suggestion (Curham et al., 2023), we have reanalysed our data to define the frequency of CD4⁺ - CD103⁻CD69⁺ T_{RM}, CD103⁺CD69⁺ T_{RM}, or total CD69⁺ T_{RM} cells in the lungs of vaccinated mice (**Fig. i**). In accordance with previous findings (Curham et al., 2023; Kumar et al., 2017), we found <10% CD4⁺T_{RM} cells in the lungs expressing CD103. Consequently, the frequency of either CD103⁺CD69⁺ or total CD69⁺ (CD103[±]CD69⁺) CD4⁺T_{RM} in J8-Lipo-DT-PHAD vaccinated versus PBS-treated mice were not statistically significant (**Fig. i**). We have included this in the results section of the manuscript. **Please refer to page 11, line 348-356.**

To clarify the relevance of our findings, which demonstrate specific accumulation of total CD69⁺ CD4⁺T_{RM} cells in J8-Lipo-DT-PHAD vaccinated mice, we have added the following statement to discussion (**pages 13-14, lines 431-439**): The specific upregulation of CD69 by tissue memory CD4⁺ and CD8⁺ T-cells in human and mice suggests that memory T-cells differentiating at tissue sites recognize distinct signals to those circulating in blood. While CD8⁺T_{RM} within mucosal site are also characterized by upregulation of the α E integrin, CD103, the role of CD103 on CD4⁺T_{RM} is less clear as many CD4⁺T_{RM} in mice and humans do not express CD103 (Coler et al., 2018; Kumar et al., 2017; Strutt et al., 2018; Turner & Farber, 2014). In agreement with these studies, we observed a low proportion of CD4⁺T_{RM}

cells expressing CD103 (<10%) in the lungs. The increased compartmentalization of CD103⁺CD69⁺CD4⁺T_{RM} cells at the lung site of mice vaccinated with J8-Lipo-DT-PHAD provides evidence that our vaccine design induces optimal T_{RM} responses.

References.

- Batzloff, M. R., Hayman, W. A., Davies, M. R., Zeng, M., Pruksakorn, S., Brandt, E. R., & Good, M. F. (2003). Protection against group A streptococcus by immunization with J8-diphtheria toxoid: contribution of J8- and diphtheria toxoid-specific antibodies to protection. *J Infect Dis*, 187(10), 1598-1608. <https://doi.org/10.1086/374800>
- Coler, R. N., Day, T. A., Ellis, R., Piazza, F. M., Beckmann, A. M., Vergara, J., Rolf, T., Lu, L., Alter, G., Hokey, D., Jayashankar, L., Walker, R., Snowden, M. A., Evans, T., Ginsberg, A., Reed, S. G., Ashman, J., Sagawa, Z. K., Tait, D., . . . The, T.-S. T. (2018). The TLR-4 agonist adjuvant, GLA-SE, improves magnitude and quality of immune responses elicited by the ID93 tuberculosis vaccine: first-in-human trial. *npj Vaccines*, 3(1), 34. <https://doi.org/10.1038/s41541-018-0057-5>
- Cunningham, M. W. (2019). Molecular Mimicry, Autoimmunity, and Infection: The Cross-Reactive Antigens of Group A Streptococci and their Sequelae. *Microbiol Spectr*, 7(4). <https://doi.org/10.1128/microbiolspec.GPP3-0045-2018>
- Curham, L. M., Mannion, J. M., Daly, C. M., Wilk, M. M., Borkner, L., Lalor, S. J., McLoughlin, R. M., & Mills, K. H. G. (2023). Bystander activation of Bordetella pertussis-induced nasal tissue-resident memory CD4 T cells confers heterologous immunity to Klebsiella pneumoniae. *Eur J Immunol*, 53(5), e2250247. <https://doi.org/10.1002/eji.202250247>
- Kumar, B. V., Ma, W., Miron, M., Granot, T., Guyer, R. S., Carpenter, D. J., Senda, T., Sun, X., Ho, S. H., Lerner, H., Friedman, A. L., Shen, Y., & Farber, D. L. (2017). Human Tissue-Resident Memory T Cells Are Defined by Core Transcriptional and Functional Signatures in Lymphoid and Mucosal Sites. *Cell Rep*, 20(12), 2921-2934. <https://doi.org/10.1016/j.celrep.2017.08.078>
- Ozberk, V., Reynolds, S., Huo, Y., Calcutt, A., Eskandari, S., Dooley, J., Mills, J. L., Rasmussen, I. S., Dietrich, J., Pandey, M., & Good, M. F. (2021). Prime-Pull Immunization with a Bivalent M-Protein and Spy-CEP Peptide Vaccine Adjuvanted with CAF[®]01 Liposomes Induces Both Mucosal and Peripheral Protection from covR/S Mutant Streptococcus pyogenes. *mBio*, 12(1). <https://doi.org/10.1128/mBio.03537-20>
- Pandey, M., & Good, M. F. (2020). A Superficial Skin Scarification Method in Mice to Mimic Streptococcus pyogenes Skin Infection in Humans. *Methods Mol Biol*, 2136, 287-301. https://doi.org/10.1007/978-1-0716-0467-0_22
- Strutt, T. M., Dhume, K., Finn, C. M., Hwang, J. H., Castonguay, C., Swain, S. L., & McKinstry, K. K. (2018). IL-15 supports the generation of protective lung-resident memory CD4 T cells. *Mucosal Immunol*, 11(3), 668-680. <https://doi.org/10.1038/mi.2017.101>
- Turner, D. L., & Farber, D. L. (2014). Mucosal resident memory CD4 T cells in protection and immunopathology. *Front Immunol*, 5, 331. <https://doi.org/10.3389/fimmu.2014.00331>
- Zaman, M., Huber, V. C., Heiden, D. L., DeHaan, K. N., Chandra, S., Erickson, D., Ozberk, V., Pandey, M., Bailly, B., Martin, G., Langshaw, E. L., Zaid, A., von Itzstein, M., & Good, M. F. (2021). Combinatorial liposomal peptide vaccine induces IgA and confers protection against influenza virus and bacterial super-infection. *Clin Transl Immunology*, 10(9), e1337. <https://doi.org/10.1002/cti2.1337>
- Zaman, M., Ozberk, V., Langshaw, E. L., McPhun, V., Powell, J. L., Phillips, Z. N., Ho, M. F., Calcutt, A., Batzloff, M. R., Toth, I., Hill, G. R., Pandey, M., & Good, M. F. (2016). Novel platform technology for modular mucosal vaccine that protects against streptococcus. *Scientific Reports*, 6(1), 39274. <https://doi.org/10.1038/srep39274>

REVIEWERS' COMMENTS

Reviewer #2 (Remarks to the Author):

One remaining comment was not addressed.

“Firstly, the pathophysiology of ARF and RHD is driven by autoimmune molecular mimicry between the *S. pyogenes* M-protein and carbohydrate cross-reacting with human-tissue.”

This sentence is in a prominent part of the paper. The authors need to make clear that this is a hypothesis/belief (even if widely held). The statement implies scientific proof and might even suggest any vaccine with M-protein or carbohydrate is wholly flawed. The authors may believe this (and may even be correct) but the review article cited does not appear to provide any proof, rather the article appears to put forward a range of hypotheses around current thinking-and indeed the article itself uses cautious language about 'streptococcal antigens' and is careful not to state that any single antigen is driving Rheumatic fever, using words like "may" and "potentially". I would suggest the authors adopt the tone of the review they cite and use terms such as 'thought to be' or 'hypothesized to be' and '*S. pyogenes* antigens'. On a very quick search I found other reviews on a similar subject state that the antigen or antigens have not been identified, for example

<https://journals.asm.org/doi/10.1128/microbiolspec.GPP3-0010-2018#T1>

or suggest alternative mechanisms

<https://www.nature.com/articles/nrcardio.2012.197>

Reviewer #3 (Remarks to the Author):

The authors have thoughtfully responded to all questions.

REVIEWERS' COMMENTS

Reviewer #2 (Remarks to the Author):

One remaining comment was not addressed.

“Firstly, the pathophysiology of ARF and RHD is driven by autoimmune molecular mimicry between the *S. pyogenes* M-protein and carbohydrate cross-reacting with human-tissue.”

This sentence is in a prominent part of the paper. The authors need to make clear that this is a hypothesis/belief (even if widely held). The statement implies scientific proof and might even suggest any vaccine with M-protein or carbohydrate is wholly flawed. The authors may believe this (and may even be correct) but the review article cited does not appear to provide any proof, rather the article appears to put forward a range of hypotheses around current thinking-and indeed the article itself uses cautious language about 'streptococcal antigens' and is careful not to state that any single antigen is driving Rheumatic fever, using words like "may" and "potentially". I would suggest the authors adopt the tone of the review they cite and use terms such as 'thought to be' or 'hypothesized to be' and '*S. pyogenes* antigens'. On a very quick search I found other reviews on a similar subject state that the antigen or antigens have not been identified, for example <https://journals.asm.org/doi/10.1128/microbiolspec.GPP3-0010-2018#T1> or suggest alternative mechanisms <https://www.nature.com/articles/nrcardio.2012.197>

We thank Reviewer #2 for their in-depth analysis of the manuscript. We have changed the following statement to reflect the Reviewer's comments. We agree that the statement implied scientific proof when in fact it is a hypothesis. Please see below the revised statement:

Firstly, the pathophysiology of ARF and RHD is thought to be driven by autoimmune molecular mimicry between *S. pyogenes* antigens cross-reacting with human-tissue.

This change has been reflected in the manuscript, please refer to page 2, lines 45-47 of the merged manuscript.

Reviewer #3 (Remarks to the Author):

The authors have thoughtfully responded to all questions.

We thank Reviewer #3 for their time and in-depth analysis of the manuscript. We believe that Reviewer #3's comments/suggestions have contributed positively to the overall manuscript. We appreciate their support.